# Tyrosine phosphorylation regulates hnRNPA2 granule protein partitioning and reduces neurodegeneration

Veronica H Ryan[1], Theodora M Perdikari[2], Mandar T Naik[3], Camillo F Saueressig[4] (iD), Jeremy Lins[4], Gregory L Dignon[5] (iD), Jeetain Mittal[5] (iD), Anne C Hart[4,*] (iD) & Nicolas L Fawzi[3,**] (iD)

## Abstract

mRNA transport in neurons requires formation of transport granules containing many protein components, and subsequent alterations in phosphorylation status can release transcripts for translation. Further, mutations in a structurally disordered domain of the transport granule protein hnRNPA2 increase its aggregation and cause hereditary proteinopathy of neurons, myocytes, and bone. We examine *in vitro* hnRNPA2 granule component phase separation, partitioning specificity, assembly/disassembly, and the link to neurodegeneration. Transport granule components hnRNPF and ch-TOG interact weakly with hnRNPA2 yet partition specifically into liquid phase droplets with the low complexity domain (LC) of hnRNPA2, but not FUS LC. *In vitro* hnRNPA2 tyrosine phosphorylation reduces hnRNPA2 phase separation, prevents partitioning of hnRNPF and ch-TOG into hnRNPA2 LC droplets, and decreases aggregation of hnRNPA2 disease variants. The expression of chimeric hnRNPA2 D290V in *Caenorhabditis elegans* results in stress-induced glutamatergic neurodegeneration; this neurodegeneration is rescued by loss of *tdp-1*, suggesting gain-of-function toxicity. The expression of Fyn, a tyrosine kinase that phosphorylates hnRNPA2, reduces neurodegeneration associated with chimeric hnRNPA2 D290V. These data suggest a model where phosphorylation alters LC interaction specificity, aggregation, and toxicity.

**Keywords** Fyn; hnRNPA2; liquid-liquid phase separation; neurodegeneration; tyrosine phosphorylation

**Subject Categories** Neuroscience; RNA Biology

**The EMBO Journal (2021) 40: e105001**

## Introduction

Cells, particularly highly polarized cells like neurons, organize their cytosol using both membrane-bound organelles and membraneless organelles (MLOs), which are condensates of RNA and proteins (Banani *et al*, 2017). Neurodegenerative diseases including amyotrophic lateral sclerosis and frontotemporal dementia (ALS/FTD) have been linked to disruption of the components and properties of MLOs, possibly through MLO stabilization by mutations in proteins capable of liquid–liquid phase separation (LLPS), a phenomenon where proteins and nucleic acids demix from the surrounding solution (Nedelsky & Taylor, 2019; Ryan & Fawzi, 2019). The relationship between stress granules, MLOs observed in cells after exposure to exogenous stress, and these ALS/FTD-associated proteins, including FUS, TDP-43, hnRNPA1, and hnRNPA2, has been a primary focus of work relating MLOs to neurodegeneration (Kim *et al*, 2013; Burke *et al*, 2015; Molliex *et al*, 2015; Patel *et al*, 2015; Conicella *et al*, 2016; Monahan *et al*, 2017; Ryan *et al*, 2018; Wang *et al*, 2018a). However, many of these disease-associated proteins are also found in physiological MLOs observed in the absence of stress, notably transport granules.

Transport granules, called neuronal granules when found in neurons, move mRNAs from the perinuclear space to sites of local translation. Importantly, many ALS/FTD-associated RNA-binding proteins that have essential nuclear functions including roles in transcription and splicing (Tollervey *et al*, 2011; Rogelj *et al*, 2012; Martinez *et al*, 2016) are also found in cytoplasmic RNA transport granules (Yasuda *et al*, 2013; Alami *et al*, 2014). Local translation is important in many cell types but is critical for myelination by oligodendrocytes and neuronal functions, including synaptic plasticity and axon guidance. Different kinds of mRNA transport granules likely exist, with distinct mRNA cargos, including neuronal β-actin mRNA transport granules containing zip-code-binding protein/ IGF2BP1 (Zhang *et al*, 2001; Elvira *et al*, 2006; Kiebler & Bassell, 2006) and hnRNPA2-containing granules transporting myelin basic

1   Neuroscience Graduate Program, Brown University, Providence, RI, USA
2   Biomedical Engineering Graduate Program, Brown University, Providence, RI, USA
3   Department of Molecular Pharmacology, Physiology, and Biotechnology, Brown University, Providence, RI, USA
4   Department of Neuroscience, Brown University, Providence, RI, USA
5   Department of Chemical and Biomolecular Engineering, Lehigh University, Bethlehem, PA, USA
    *Corresponding author. Tel: +1 401 863 2822; E-mail: anne_hart@brown.edu
    **Corresponding author. Tel: +1 401 863 5232; E-mail: nicolas_fawzi@brown.edu

protein mRNA in oligodendrocytes or Arc, neurogranin, and αCamKII mRNAs in neurons (Ainger *et al*, 1993; Brumwell *et al*, 2002; Shan *et al*, 2003; Gao *et al*, 2008). Several other protein components of hnRNPA2-containing transport granules have been identified, including hnRNPF, hnRNPAB, hnRNPE1, hnRNPK, and the microtubule-associated protein ch-TOG (CKAP5) (Kosturko *et al*, 2005; Kosturko *et al*, 2006; Francone *et al*, 2007; Laursen *et al*, 2011; Raju *et al*, 2011; White *et al*, 2012; Torvund-Jensen *et al*, 2014). Importantly, hnRNPA2 transport granules appear to contain a specific set of proteins and exclude related proteins associated with other cytoplasmic granules (e.g., other RNA-binding proteins found in stress granules), yet the mechanisms of granule component specificity remain unclear. Some of these protein interactions with hnRNPA2 are RNA-dependent (Laursen *et al*, 2011; Raju *et al*, 2011; Torvund-Jensen *et al*, 2014) but hnRNPF and ch-TOG both directly bind to hnRNPA2 (Kosturko *et al*, 2005; White *et al*, 2012; Falkenberg *et al*, 2017). Interactions within the granules are regulated; mRNA is released for local translation in processes when hnRNPA2 and hnRNPF are locally phosphorylated by the tyrosine kinase Fyn (White *et al*, 2008; White *et al*, 2012). Yet, the molecular basis for both hnRNPA2-hnRNPF and hnRNPA2-TOG interactions and their disruption by phosphorylation remains unknown.

hnRNPA2 mutations cause multisystem proteinopathy (MSP), a degenerative disease with clinical features of ALS/FTD, inclusion body myopathy, and Paget's disease of bone (PDB) (mutation: D290V) (Kim *et al*, 2013) as well as PDB alone in a separate family (mutation: P298L) (Qi *et al*, 2017). These disease mutations drive aggregation of the protein *in vitro* (Kim *et al*, 2013; Ryan *et al*, 2018). The D290V mutation also enhances hnRNPA2 stress granule localization (Martinez *et al*, 2016) even in the absence of stress (Kim *et al*, 2013), induces aggregation in the cytoplasm (Kim *et al*, 2013) and nucleus (Martinez *et al*, 2016), and results in abnormal splicing changes and decreased survival in neuronal culture (Martinez *et al*, 2016). Therefore, the mechanistic interactions leading to hnRNPA2 function and dysfunction provide an interesting model for a large class of RNA-binding proteins mutated in disease (King *et al*, 2012). hnRNPA2 contains two RNA recognition motifs (RRMs) and a glycine-rich low complexity (LC) domain, which is necessary and sufficient for LLPS (Ryan *et al*, 2018). Importantly, both mutations are located in the aggregation-prone "prion-like" LC domain (named for resemblance in sequence composition to the polar-residue-rich domains found in yeast prion proteins) (King *et al*, 2012). Given that transport granule proteins are thought to interact directly with hnRNPA2 LC (Falkenberg *et al*, 2017) and that the LC gives hnRNPA2 the ability to undergo LLPS, we set out to determine the molecular basis for interactions of hnRNPA2 LC with hnRNPF prion-like domain (PLD) and TOG domain 1 (D1) by evaluating their ability to specifically co-phase separate into *in vitro* models of reconstituted multicomponent hnRNPA2 granules. We chose these domains to probe the molecular interactions between the disordered LC or PLD domains and a single domain of TOG, as the disordered domains are thought to be protein–protein interaction domains and contribute to LLPS and granule formation. Furthermore, as post-translational modifications (PTMs) alter phase separation and aggregation of RNA-binding proteins (Nott *et al*, 2015; Monahan *et al*, 2017; Ryan *et al*, 2018; Wang *et al*, 2018a), we tested the hypothesis that tyrosine phosphorylation disrupts the interactions between granule components as a possible mechanism

for granule dissociation and prevents protein aggregation *in vitro*. We also hypothesized that hnRNPA2 mutation could induce neurodegeneration in an animal model and that promoting tyrosine phosphorylation could prevent toxicity. Here, we used nuclear magnetic resonance (NMR) spectroscopy, molecular simulation, *in vitro* phase separation assays, and a novel *C. elegans* model to map the protein–protein interactions between components of hnRNPA2-containing transport granule assembly, probe a potential mechanism of physiological disassembly, and assess strategies to prevent hnRNPA2-associated neurodegeneration.

## Results

### hnRNPA2 arginine residues are required for *in vitro* interaction with transport granule component hnRNPF

hnRNPA2 was previously shown to undergo LLPS (Ryan *et al*, 2018) and interact with other protein components of myelin basic protein mRNA transport granules, including hnRNPF (White *et al*, 2012). Importantly, hnRNPF interacts directly with hnRNPA2 protein in transport granules (White *et al*, 2012), but the biophysical details of this interaction between these two prion-like domain containing proteins remain unclear. Here, we sought to reconstitute and structurally characterize this interaction using recombinant full-length proteins and their isolated domains in order to probe the unexplored interactions between these proteins that occur inside granules.

First, we tested which domains mediate the hnRNPA2-hnRNPF interaction. Using purified recombinant hnRNPA2 low complexity domain (LC, residues 190–341) and full length (FL) with hnRNPF prion-like domain (PLD, resides 365–415), and FL (Appendix Fig S1A), we asked if fluorescently tagged hnRNPF partitions into hnRNPA2 droplets. hnRNPF (PLD or FL) did not undergo LLPS alone in the conditions and concentrations required for LLPS of hnRNPA2 LC or FL (Fig 1A, Appendix Fig S1B and C). However, hnRNPF PLD partitioned into hnRNPA2 LC (Fig 1A) and hnRNPA2 FL (Appendix Fig S1B) droplets and was equally distributed throughout the droplets. hnRNPF FL partitioned into both hnRNPA2 LC and hnRNPA2 FL droplets (Appendix Fig S1B and C). The PLD of hnRNPF was not essential for partitioning into hnRNPA2 droplets, as hnRNPF lacking the PLD (hnRNPF ΔPLD, residues 1–364) partitioned into hnRNPA2 FL droplets (Appendix Fig S1B). We attempted to test whether the hnRNPF PLD is necessary for partitioning into hnRNPA2 LC droplets and while some co-localization was observed (Appendix Fig S1C), hnRNPF ΔPLD seemed to aggregate, possibly because the hnRNPA2 LC LLPS assay conditions required crossing the hnRNPF ΔPLD predicted isoelectric point (pI), where proteins are likely to self-assemble. Taken together, these data suggest that the PLD of hnRNPF avidly partitions into hnRNPA2 droplets.

Next, we focused on delineating the molecular basis for the interaction between hnRNPA2 LC and hnRNPF PLD. To visualize hnRNPF PLD with residue-by-residue resolution, we performed multidimensional NMR spectroscopy to observe each backbone amide position. From the narrow chemical shift dispersion and secondary shifts, we conclude that hnRNPF PLD is primarily disordered (Appendix Fig S2A and B). Then, we attempted to localize the sites of interactions between hnRNPA2 LC and hnRNPF PLD using

NMR titrations. These experiments were performed at low salt conditions where hnRNPA2 LC alone does not phase separate (mimicking monomeric interactions) as NMR signal is attenuated due to LLPS into viscous, sedimenting droplets when hnRNPA2 LC is placed in the presence of salt. We found that addition of hnRNPF PLD reduces the signal intensity of hnRNPA2 LC; the converse is also true: Addition of hnRNPA2 LC reduces the signal intensity of hnRNPF PLD (Appendix Fig S2C and D). These findings are reminiscent of how Fyn-SH3, a folded domain that does not phase separate on its own, reduces hnRNPA2 LC signal intensity by inducing LLPS

of hnRNPA2 LC (Amaya *et al*, 2018). However, we performed microscopy at the NMR conditions and found that hnRNPF PLD aggregates in these low salt conditions and these aggregates remain in the presence of hnRNPA2 LC (Appendix Fig S2E). Therefore, the decrease in hnRNPA2 LC NMR signal intensity in the presence of hnRNPF PLD is likely attributable to co-aggregation or interaction of hnRNPA2 LC with hnRNPF PLD, either of which are consistent with interaction between these domains. The NMR titrations showed signal attenuation across the entire length of hnRNPF PLD and hnRNPA2 LC, providing no evidence for specific residues or residue

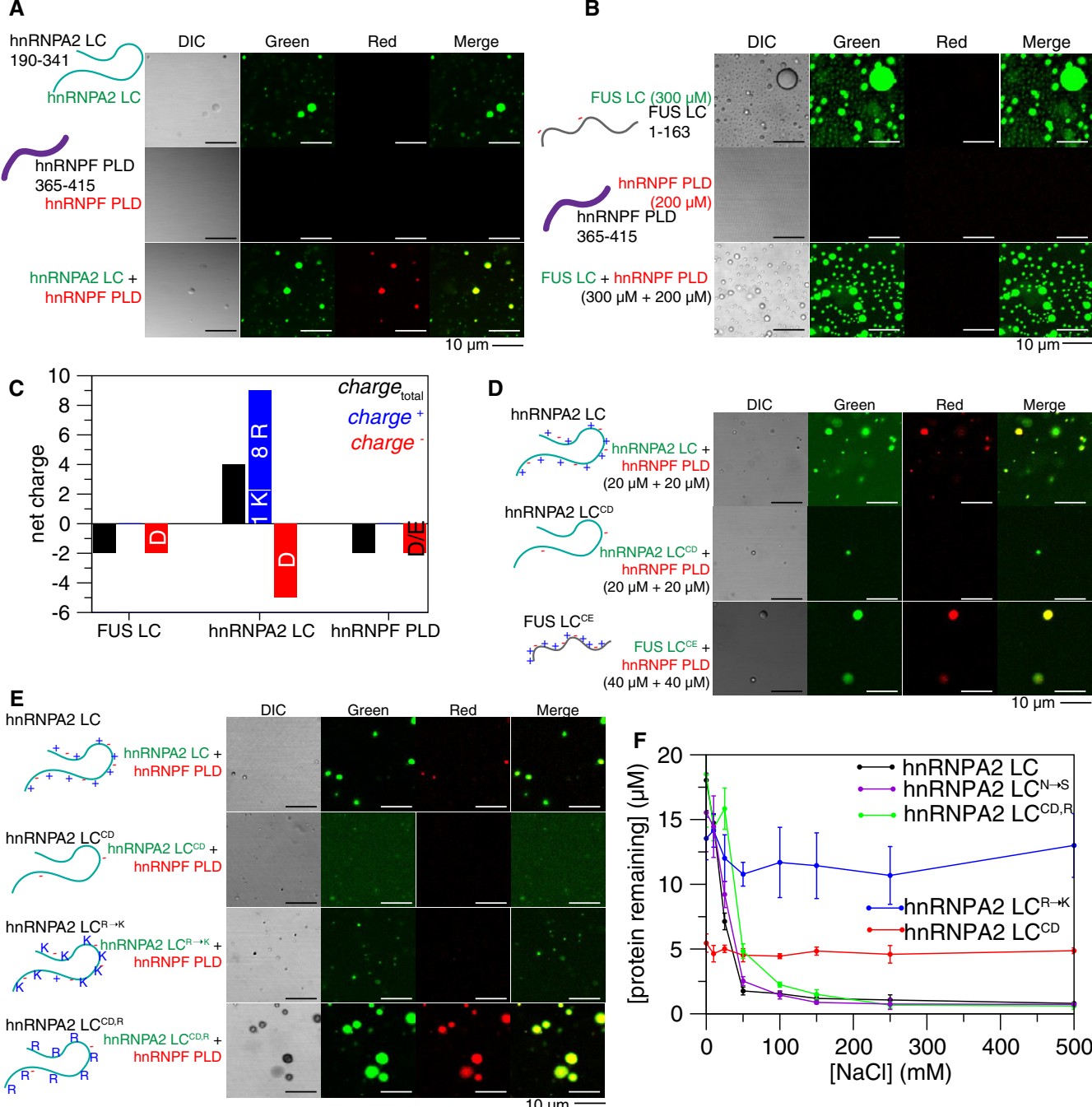

**Figure 1.**

**Figure 1.  Arginine in hnRNPA2 LC is required for the interaction with granule component hnRNPF PLD. See also Appendix Figs S1–S3.**

A  hnRNPA2 LC (AlexaFluor 488-tagged, green) undergoes LLPS, while hnRNPF PLD (AlexaFluor 555-tagged, red) does not. However, hnRNPF PLD partitions into hnRNPA2 LC droplets when mixed at a 1:1 ratio. Conditions: 20 μM indicated protein (~1% fluorescently tagged), 20 mM MES pH 5.5, 50 mM NaCl, 150 mM urea. Scale bar: 10 μm.

B  At 300 μM, FUS LC (AlexaFluor488-tagged, green) undergoes LLPS, but at 200 μM hnRNPF PLD (AlexaFluor555-tagged, red) still does not undergo LLPS. When mixed at 300 μM FUS LC and 200 μM hnRNPF PLD, hnRNPF PLD does not partition into FUS LC droplets. Conditions: 300 μM FUS and 200 μM hnRNPF PLD (~1% fluorescently tagged), 20 mM MES pH 5.5, 150 mM NaCl, 150 mM urea. Scale bar: 10 μm.

C  While FUS LC and hnRNPF PLD both have a small negative predicted net charge at neutral pH, hnRNPA2 LC has a predicted + 4 net positive charge, due to the 9 positively charged residues (8 arginine, 1 lysine) and 5 negatively charged residues.

D  Removal of the charged residues from hnRNPA2 LC (hnRNPA2 LC$^{CD}$) prevents partitioning of hnRNPF PLD into the hnRNPA2 LC phase. Addition of hnRNPA2 LC-like charged residue patterning to FUS LC (FUS LC$^{CE}$) allows the partitioning of hnRNPF PLD at 40 μM. Conditions: protein concentration indicated next to image (20 μM hnRNPA2 LC and hnRNPA2 LC$^{CD}$, 40 μM FUS LC$^{CE}$, hnRNPF PLD concentration matches other protein in mixture (either 20 or 40 μM)) (all ~ 1% fluorescently tagged), 20 mM MES pH 5.5 50 mM NaCl, 150 mM urea. Scale bar: 10 μm.

E  Substitution of all arginines in hnRNPA2 LC with lysine prevents the partitioning of hnRNPF PLD into hnRNPA2 LC$^{R→K}$ droplets. Removing all charged residues except for arginine from hnRNPA2 LC (hnRNPA2 LC$^{CD,R}$) allows partitioning of hnRNPF PLD into droplets, indicating arginine in hnRNPA2 LC is required and necessary for hnRNPF partitioning. hnRNPA2 LC$^{R→K}$ does not phase separate much as hnRNPA2 LC at these conditions, see Appendix Fig S1J for quantification of phase separation of variants. Conditions: 20 μM proteins, 20 mM MES pH 5.5 50 mM NaCl, 150 mM urea. Scale bar: 10 μm.

F  Quantification of phase separation of hnRNPA2 LC constructs used to determine the residue types important for hnRNPF PLD partitioning. hnRNPA2 LC$^{N→S}$ (purple) has similar phase separation to hnRNPA2 LC. hnRNPA2 LC$^{CD}$ (red) is consistently phase separated with ~ 5 μM protein remaining in the supernatant at all salt conditions tested. Adding back arginines to hnRNPA2 LC no charge (hnRNPA2 LC$^{CD,R}$, green) brings phase separation as a function of salt to similar levels as hnRNPA2 LC. Changing all the arginine residues to lysine (removing the π-character but maintaining positive charge, hnRNPA2 LC$^{R→K}$) also removes the salt dependence of phase separation but has reduced phase separation overall. Conditions: 20 μM of each protein, pH 5.5 MES, NaCl concentration as indicated, 25° C. Error bars are standard deviation of three replicates.

types in hnRNPA2 LC interactions with hnRNPF PLD (Appendix Fig S2C and D), consistent with the repetitive, low complexity sequence of these domains.

An important question in the field of MLOs is the origin of specificity of granule partitioning. Although no specific sites of interaction were observed by NMR, we wondered if the sequences of hnRNPA2 LC and hnRNPF PLD could encode any specificity in partitioning. Therefore, we examined specificity by testing if hnRNPF PLD could co-phase separate with the low complexity domain of the RNA-binding protein FUS. FUS is also found in stress granules but has not been found in hnRNPA2-myelin basic protein mRNA granules. Importantly, FUS LC has different amino acid composition than that of hnRNPA2. However, we found that FUS LC and FUS LC 12E, a phosphomimetic form (incorporating 12 serine to glutamate (E) substitutions) of FUS that does not undergo of FUS that does not undergo LLPS (Monahan *et al*, 2017), are both capable of partitioning into hnRNPA2 LC (Appendix Fig S3A). Interestingly, when hnRNPF PLD was mixed with FUS LC, it did not partition into FUS LC droplets (Fig 1B), demonstrating specificity of the hnRNPA2-hnRNPF partitioning. We hypothesized that charged residues might underlie this specificity, as both FUS LC and hnRNPF PLD are free of positively charged resides and are slightly negatively charged, while hnRNPA2 LC has a net positive charge (Fig 1C). To test the role of charged residues in hnRNPA2 LC specifying the interaction with hnRNPF, we changed almost all the charged residues from hnRNPA2 LC to serine or glutamine resulting in a "FUS LC-like" charged residue pattern (which we termed hnRNPA2 LC$^{CD}$, for "charge depleted") or changed serine/glutamine residues in FUS to charged residues to give it "hnRNPA2 LC-like" charged residue pattern including net charge, identity of charged residues, and approximate spacing between charged residues (FUS LC$^{CE}$, for "charge enhanced"), and examined hnRNPF PLD partitioning into these charged residue variants. hnRNPF PLD did not partition into hnRNPA2 LC$^{CD}$ but did partition into FUS LC$^{CE}$ (Fig 1D, Appendix Fig S3B), consistent with our hypothesis.

Given that arginine contacts with aromatic residues are important for phase separation in hnRNPA2 (Ryan *et al*, 2018) and other

proteins (Vernon *et al*, 2018; Wang *et al*, 2018b), we hypothesized that partitioning of hnRNPF PLD into hnRNPA2 LC droplets required arginine in hnRNPA2 LC and tyrosine in hnRNPF PLD. Therefore, we changed hnRNPF PLD tyrosine to serine (hnRNPF PLD$^{Y→S}$), hnRNPA2 LC arginine to lysine (hnRNPA2 LC$^{R→K}$), and also tested a form of hnRNPA2 LC with arginine residues retained but otherwise "FUS LC-like" charged residue depleted patterning (hnRNPA2 LC$^{CD,R}$). First, we found that hnRNPF PLD$^{Y→S}$ still partitioned into hnRNPA2 LC droplets (Appendix Fig S3C), implying that tyrosine in hnRNPF PLD is not the only residue type contributing to the interaction between hnRNPA2 LC and hnRNPF PLD. Second, we found that replacing arginine with lysine in hnRNPA2 LC to maintain net charge and charge patterning (hnRNPA2 LC$^{R→K}$) prevented partitioning of hnRNPF PLD or hnRNPF PLD$^{Y→S}$, suggesting that arginine-specific contacts and not just positive charge determine partitioning (Fig 1E, Appendix Fig S3C). Third, we found that adding back arginine to hnRNPA2 LC$^{CD}$ allowed partitioning of hnRNPF PLD and hnRNPF PLD$^{Y→S}$ (Fig 1E, Appendix Fig S3C). Inconveniently, FUS LC$^{CE,R→K}$ and FUS LC$^R$ did not undergo LLPS at the same conditions as hnRNPA2 LC, so we could not perform the complementary experiments testing hnRNPF PLD partitioning into modified FUS constructs (Appendix Fig S3C). We also tested the role of asparagine residues in hnRNPA2 LC by changing asparagine to serine (hnRNPA2 LC$^{N→S}$) and the effect of serine residues in hnRNPF PLD by changing serine to alanine (hnRNPF PLD$^{S→A}$), but these changes did not qualitatively alter partitioning (Appendix Fig S3C). Interestingly, both the hnRNPA2 LC$^{CD}$ and hnRNPA2 LC$^{R→K}$ substantially altered LLPS of hnRNPA2 LC, while hnRNPA2 LC$^{N→S}$ and hnRNPA2 LC$^{CD,R}$ had similar LLPS to hnRNPA2 LC (Fig 1F). Finally, we also tested the ability of hnRNPF PLD and FL to partition into FUS FL droplets. We found that both hnRNPF PLD and FL are able to partition into FUS FL droplets (Appendix Fig S3D), which is expected as FUS FL has three arginine-rich domains outside of the N-terminal LC. Therefore, though these results suggest some specificity in the interaction of disordered domains arises directly from their sequence, additional factors such as RNA-binding specificity

(Helder *et al*, 2016) mediated by the RNA-binding domains in the full-length proteins likely contribute to granule component sorting *in vivo*. We conclude that arginine residues in hnRNPA2 LC are critical to specify co-phase separation with hnRNPF PLD.

### hnRNPA2 LC interacts with TOG D1 weakly *in vitro* through its disordered loops and helical face

Similar to hnRNPF, ch-TOG was previously shown to interact with hnRNPA2 in myelin basic protein transport granules via contacts with hnRNPA2 LC (Kosturko *et al*, 2005; Francone *et al*, 2007; Falkenberg *et al*, 2017). Given our results with hnRNPF, we hypothesized that one domain of ch-TOG, TOG D1 (residues 1–250) (Appendix Fig S4A), would also partition into hnRNPA2 droplets. We used a single domain of ch-TOG as the individual domains are highly similar in sequence and structure and each isolated domain was previously shown to interact similarly with hnRNPA2 at nanomolar affinity (Falkenberg *et al*, 2017). As expected for a single globular domain, recombinant TOG D1 did not undergo LLPS in the micromolar concentration range or buffers tested, but fluorescently tagged TOG D1 partitioned into hnRNPA2 LC and FL droplets (Fig 2A, Appendix Fig S4B). To map the interactions between hnRNPA2 LC and TOG D1, we performed NMR resonance assignment experiments on TOG D1 and confirmed its primarily α-helical secondary structure (Appendix Fig S4C and D). Surprisingly, when we performed NMR titrations with hnRNPA2 LC and TOG D1, we found no significant interaction between hnRNPA2 LC and TOG D1 in the dispersed (not phase-separated) state using this technique (Appendix Fig S4E and F), counter to our expectations. The lack of evidence for tight binding suggests that the dissociation constant for the interaction between these two proteins is likely in the micromolar to millimolar range, weaker than previously reported (which we attribute to possible artifacts arising from LLPS or aggregation, see Discussion).

Similar to our results with hnRNPF, we observed that the co-phase separation of hnRNPA2 LC and TOG D1 is specific; TOG D1 partitioned into hnRNPA2 LC but not FUS LC droplets (Fig 2B). However, as with hnRNPF, TOG D1 is able to partition into FUS FL droplets (Appendix Fig S5A). We hypothesized that because TOG D1 has a charged surface (Appendix Fig S5B), the charge variants of hnRNPA2 LC and FUS LC would also alter TOG D1 partitioning. Indeed, TOG D1 partitioned into FUS LC$^{CE}$ droplets, yet it also partitioned into hnRNPA2 LC$^{CD}$ droplets (Appendix Fig S5C). This result is different than what we observed for hnRNPF PLD and indicates that charged residues contribute to the specificity of partitioning, but other interactions contribute as well. To begin to elucidate these interactions, we turned to NMR techniques that provide position-specific information on weak, transient interactions. High resolution transverse relaxation optimized (TROSY-based) paramagnetic relaxation enhancement (PRE) experiments on mixtures of NMR invisible hnRNPA2 LC tagged with a small (~120 Da) paramagnetic label at a single engineered cysteine site (either S285C or S329C) (Ryan *et al*, 2018) and NMR-visible ($^2$H $^{15}$N) TOG D1 revealed weak interactions, consistent with a dissociation constant weaker than 100 µM, across several sites on TOG D1 (Appendix Fig S5D). These interactions suggest that the region of hnRNPA2 LC bearing the paramagnetic tag comes in close proximity with particular parts of the TOG D1 surface. We then mapped the PRE values on a homology model of TOG D1 and found

that they localized to disordered loops and helix faces of TOG D1 (Fig 2C and D). Upon sorting PRE values ≥ 0.5 s$^{-1}$ by residue type, enhancement is most often observed at non-polar positions (28/76 residues with PREs) and charged residues (27/76). However, non-polar amino acids are the most prevalent type in TOG D1 (70/165 assigned residues); after normalizing to the number of assignable residues of that type, residues with PREs are enriched in polar residues (22/40 assigned), suggesting that polar contacts on the surface of TOG D1 may be important for the interaction. Interestingly, PREs arising from hnRNPA2 LC S285C, labeled near the aggregation-prone PLD segment within hnRNPA2 LC, showed larger PREs than hnRNPA2 LC labeled at S329C in the glycine-rich tail, indicating a stronger interaction with the highly conserved PLD sub-region (Kim *et al*, 2013). hnRNPA2 LC self-interactions are also weaker when probed with the label at S329C than at S285C (Ryan *et al*, 2018), so it is possible that hnRNPA2 LC interacts with TOG D1 using similar polar-residue contacts as hnRNPA2 LC uses for self-interactions.

Given that hnRNPA2, hnRNPF, and TOG are found in granules simultaneously in cells, we hypothesized that all three could be found in our *in vitro* droplets simultaneously as well. We performed partitioning experiments and found that hnRNPF PLD and TOG D1 were able to partition with hnRNPA2 LC (Fig 2E), though consistent with our results above hnRNPA2 LC appears to aggregate instead of for a liquid droplet in the presence of hnRNPF PLD. Further, we found that hnRNPF FL and TOG D1 co-partition into hnRNPA2 LC liquid droplets (Fig 2E, Appendix Fig S6A). Additionally, hnRNPF PLD and TOG D1 as well as hnRNPF FL and TOG D1 were able to co-partition into hnRNPA2 FL droplets at the same time (Appendix Fig S6B). These results suggest that these *in vitro* models of granules are capable of reconstituting the multicomponent nature of hnRNPA2 granules where simultaneous interaction with several partners is likely to occur.

### Tyrosine phosphorylation of hnRNPA2 LC alters *in vitro* LLPS and prevents partitioning of granule components

Because hnRNPA2 LC is a known target of tyrosine phosphorylation at 13 sites or more (Hornbeck *et al*, 2015) and tyrosine phosphorylation was previously shown to release mRNA from hnRNPA2 granules for translation in cells (White *et al*, 2008), we hypothesized that tyrosine phosphorylation of hnRNPA2 LC would alter phase separation propensity, as we found previously for serine/threonine phosphorylation of FUS LC (Monahan *et al*, 2017). We generated and purified recombinant partially tyrosine phosphorylated (pY) hnRNPA2 LC with approximately 4 to 8 phosphorylated tyrosine residues (of 17 total; by mass spectrometry) (Appendix Fig S7A). We found that pY hnRNPA2 LC undergoes LLPS with an inverse salt dependence compared to unphosphorylated hnRNPA2 LC. Specifically, pY hnRNPA2 LC phase separated more at low salt conditions and less at high salt conditions (Fig 3A). The high phase separation propensity at low salt conditions may be because some of the phosphorylated forms are at or near their pI. We also performed microscopy experiments with pY hnRNPA2 using a subset of conditions and always observed droplets (Fig 3B). To determine whether the altered LLPS of pY hnRNPA2 LC is a unique feature of phosphotyrosine or is due to increased net negative charge of pY, we tested if hnRNPA2 LC with phosphoserine mimics could also alter LLPS as a function of salt concentration. We generated two phosphomimic

constructs, hnRNPA2 LC[5E], where 5 serine residues are changed to glutamate, and hnRNPA2 LC[12E], where all twelve serine residues are changed to glutamate (hnRNPA2 LC contains no threonine). At physiological ionic strength (150 mM NaCl) and above, we find

that, compared to unmodified hnRNPA2 LC, hnRNPA2 LC[5E] shows a modest reduction in LLPS while hnRNPA2 LC[12E] shows substantially less LLPS (Appendix Fig S7B), consistent with introduction of negatively charged residues disrupting LLPS. Interestingly, at low

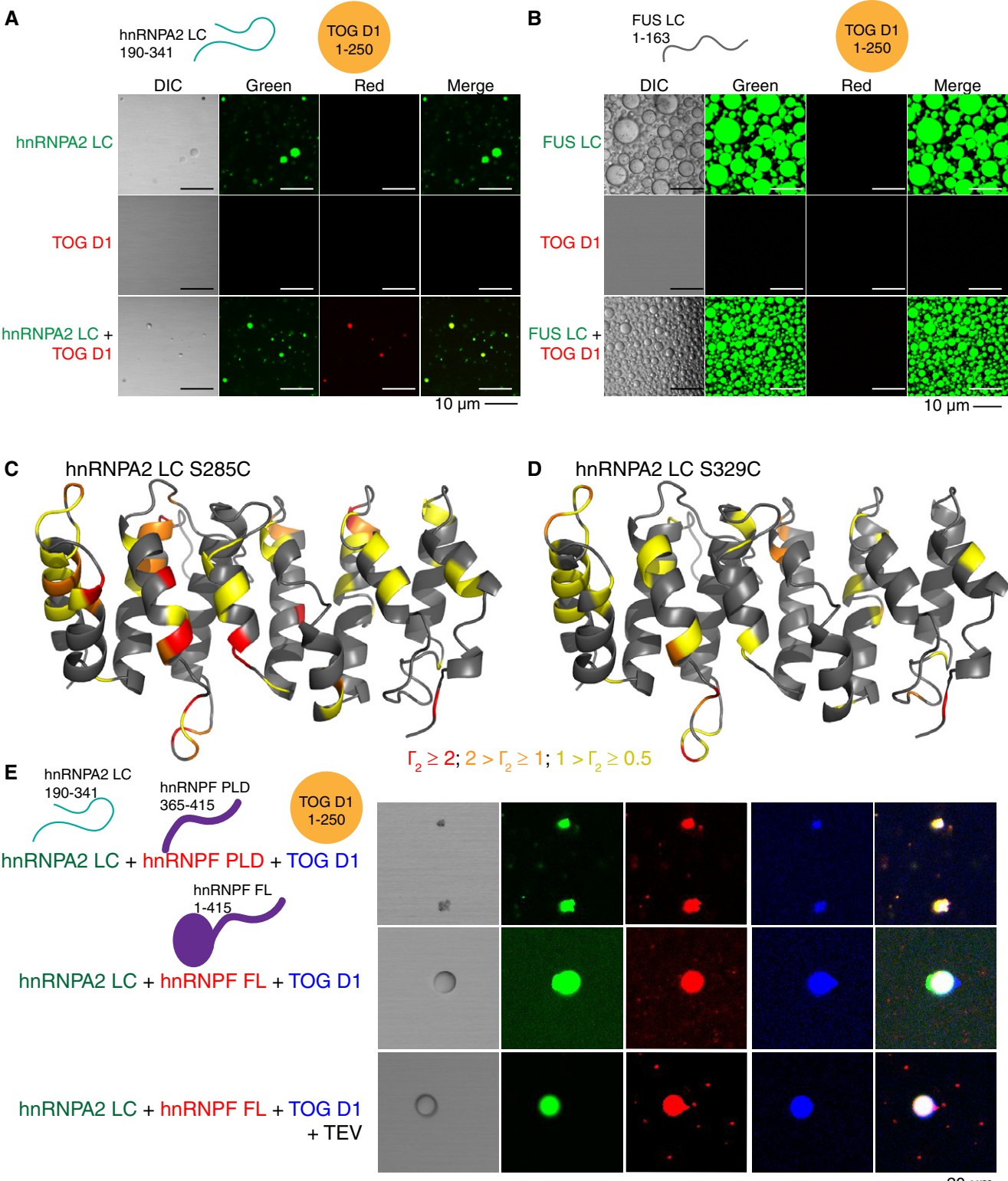

**Figure 2.**

**Figure 2.   Transport granule component TOG D1 interacts weakly with hnRNPA2 LC. See also Appendix Figs S4–S6.**

A       AlexaFluor 488-tagged (green) hnRNPA2 LC undergoes LLPS, while AlexaFluor 555-tagged (red) TOG D1 does not. However, TOG D1 partitions into hnRNPA2 LC droplets when mixed at a 1:1 ratio. Conditions: 20 μM indicated protein (~1% fluorescently tagged), 20 mM MES pH 5.5, 50 mM NaCl, 150 mM urea. Scale bar: 10 μm. hnRNPA2 LC control duplicated from Fig 1A as hnRNPF PLD and TOG D1 samples were made concurrently.

B       Similar to hnRNPF PLD, AlexaFluor 555-tagged TOG D1 does not undergo LLPS at 300 μM or partition into AlexaFluor488-tagged FUS LC droplets with both proteins at 300 μM. Conditions: 300 μM proteins (~1% fluorescently tagged), 20 mM MES pH 5.5 150 mM NaCl, 150 mM urea. Scale bar: 10 μm.

C, D  TOG D1 homology structure with $\Gamma_2$ values from PRE experiments for hnRNPA2 LC (C) S285C and (D) S329C. Amino acids are colored based on $\Gamma_2$ value: Red corresponds to $\Gamma_2 > 2$, orange to $2 > \Gamma_2 > 1$, yellow to $1 > \Gamma_2 > 0.5$.

E       AlexaFluor 555-tagged hnRNPF PLD or FL and AlexaFluor 405-tagged TOG D1 partition simultaneously into AlexaFluor 488-tagged hnRNPA2 LC droplets. Conditions: 20 μM of each indicated protein (~1% fluorescently tagged), 20 mM MES pH 5.5, 50 mM NaCl, 150 mM urea. Scale bar: 20 μm.

salt conditions (0 mM NaCl), hnRNPA2 LC[5E] phase separates, consistent with this variant being at its pI. To identify the interactions underlying LLPS propensity for pY and serine phosphomimic constructs, we performed NMR titrations of free amino acids (pY, Y, pS, R) into hnRNPA2 LC. Due to the high propensity of hnRNPA2 LC to phase separate, we were only able to perform these analyses at 0 mM NaCl. We found that while tyrosine, arginine, and phosphoserine do not interact strongly with hnRNPA2 LC, addition of phosphotyrosine induced LLPS of hnRNPA2 LC and seems to preferentially interact with arginine in hnRNPA2 LC (Appendix Fig S7C and D). In contrast, hnRNPF PLD shows no decrease in signal intensity with free tyrosine, phosphotyrosine, or arginine (Appendix Fig S7E). These results suggest that, at low salt conditions, arginine to phosphotyrosine interactions may be important for pY hnRNPA2 LC LLPS. However, when tested in conditions where hnRNPA2 LC phase separates (50 mM NaCl), free pY, Y, and pS were unable to prevent partitioning of hnRNPF PLD or TOG D1 (Appendix Fig S7F), suggesting that these pY-R interactions are insufficient to fully recapitulate the interactions between proteins.

We next asked if tyrosine phosphorylation of hnRNPA2 LC could alter partitioning of hnRNPF or TOG D1. At 50 mM NaCl, both hnRNPA2 LC and pY hnRNPA2 LC undergo LLPS. TOG D1 and hnRNPF PLD each partitioned into hnRNPA2 LC droplets, but interestingly, neither partitioned into pY hnRNPA2 LC (Fig 3C, Appendix Fig S8A). Additionally, hnRNPF FL failed to partition into pY hnRNPA2 LC droplets (Appendix Fig S8A). In contrast, the serine phosphomimic constructs hnRNPA2 LC[5E] and hnRNPA2 LC[12E] did not prevent partitioning of hnRNPF PLD or TOG D1 (Fig 3D, Appendix Fig S8B), suggesting that tyrosine phosphorylation may have a unique role in disrupting hnRNPA2 LC interactions with other hnRNPA2 granule components and that the increased net negative charge arising from phosphorylation is not exclusively responsible for the specificity of interactions with hnRNPA2 LC. As arginine was shown to be important for both hnRNPF PLD partitioning and phase separation of phosphotyrosine hnRNPA2 LC, it is possible that hnRNPA2 self-interactions between phosphotyrosine and arginine outcompete the weak interactions with hnRNPF PLD, thus preventing its partitioning. Combined, these results provide molecular detail to a previously suggested model in which hnRNPA2 LC tyrosine phosphorylation may play a critical role in regulating LLPS and granule formation/disassociation (White *et al*, 2008; Muller *et al*, 2013).

## Tyrosine phosphorylation of hnRNPA2 FL prevents *in vitro* aggregation of disease mutants

Phosphorylation can reduce the aggregation of disease-associated proteins (Monahan *et al*, 2017). Therefore, we hypothesized that aggregation of hnRNPA2 FL disease variants might be altered by

tyrosine phosphorylation. We generated recombinant pY hnRNPA2 FL using the same strategy described for hnRNPA2 LC. After cleavage of a C-terminal maltose-binding protein solubility tag, hnRNPA2 FL WT undergoes robust LLPS within 30 min, but pY hnRNPA2 FL WT does not phase separate even after 2 h (Fig 4A). Consistent with our previous work (Ryan *et al*, 2018), hnRNPA2 FL D290V forms aggregates after 30 min; these aggregates increase in apparent size and number by 2 h. In contrast, pY hnRNPA2 FL D290V only formed small droplets after 30 min that do not appear to substantially increase in size or form amorphous aggregates in the 2-h assay period (Fig 4A). Similar to WT, hnRNPA2 FL P298L showed LLPS after 30 min but formed aggregates after 2 h. Yet, pY hnRNPA2 FL P298L does not undergo LLPS at all, even 2 h after maltose-binding protein cleavage (Fig 4A), like the phosphorylated WT. We conclude that disease variants of hnRNPA2 aggregate over time under these conditions, but that tyrosine phosphorylation can decrease LLPS and prevent aggregation.

Given these striking differences in phase separation and aggregation, we sought to test if phosphorylation alters the prion-like character of hnRNPA2. We performed "yeast prion-like" amino acid composition (PLAAC) analysis (Lancaster *et al*, 2014) on hnRNPA2 FL and four alternate forms: 1) hnRNPA2 with 5 serine to glutamate mutations in the LC to mimic 5 serine phosphorylations, 2) 12 serine to glutamine mutations to mimic phosphorylation of all serine residues of hnRNPA2, 3) 17 tyrosine to glutamate mutations to attempt to mimic tyrosine phosphorylation of all tyrosine residues in hnRNPA2 LC, and 4) 17 tyrosine to glutamine mutations to determine the effect of removing the aromatic ring of tyrosine without altering protein charge (because glutamine and glutamate are isosteric, this comparison isolates the importance of the introduction of a negatively charged residue at tyrosine positions as a control for the Y→E phosphomimic). The serine phosphomimics did reduce the prion-like character of hnRNPA2 slightly (as compared to non-phosphorylated WT hnRNPA2 FL), but the tyrosine phosphomimic drastically reduced the prion-like character of hnRNPA2 FL, while the tyrosine to glutamine increased prion-like character (Fig 4B). As substitutions to charged glutamates are known to reduce the prion-like character of proteins in PLAAC analysis, we also performed ZipperDB analysis (Thompson *et al*, 2006) to see whether tyrosine phosphomimic reduces the predicted stability of steric zipper amyloids which are readily formed by the PLD of mutant forms of hnRNPA2 LC (Kim *et al*, 2013; Ryan *et al*, 2018). Indeed, tyrosine to glutamate mutations, but not serine to glutamate or tyrosine to glutamine mutations, reduced the predicted favorability of steric zipper formation such that the computed free energy no longer crosses the threshold predicting amyloid formation (Fig S9A). As prion-like character has been suggested to tune the ability to undergo phase separation (Franzmann & Alberti, 2019), the

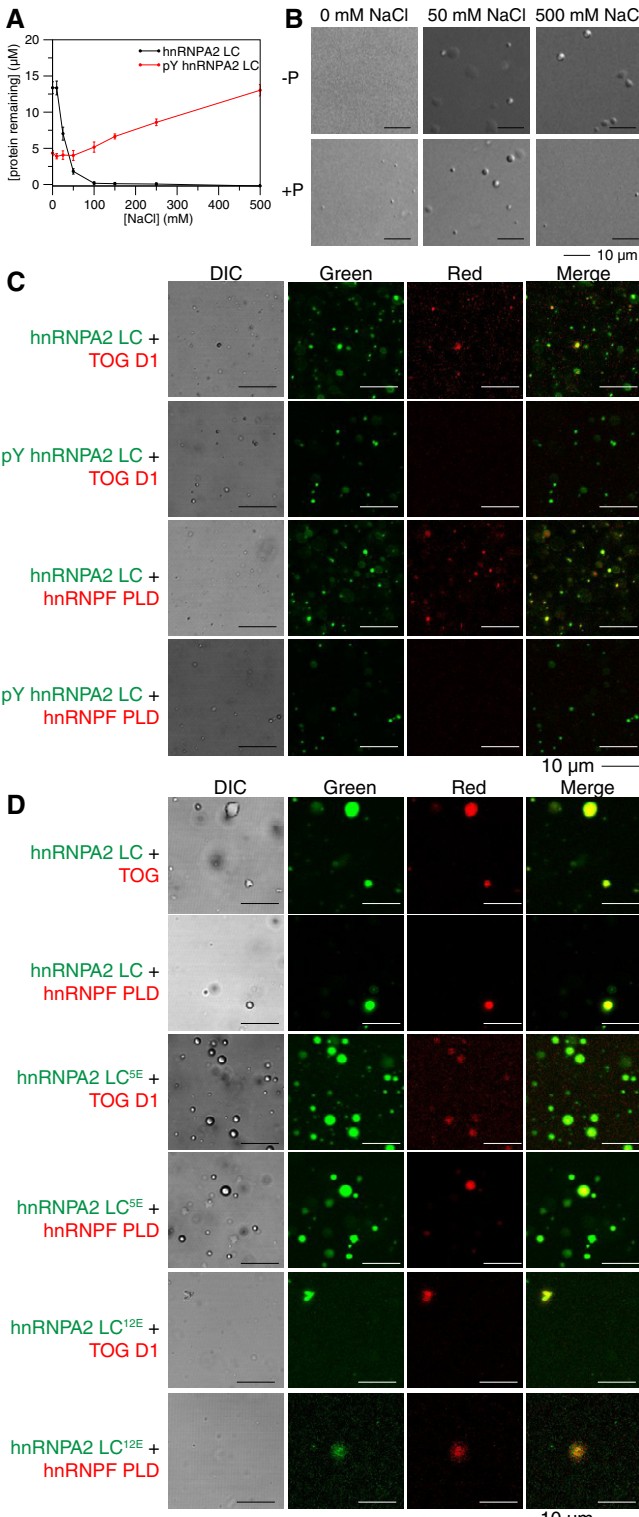

Figure 3.  Tyrosine phosphorylation of hnRNPA2 LC alters LLPS and prevents partitioning of hnRNPF PLD or TOG D1. See also Appendix Figs S7 and S8.

A  Concentration of protein remaining in the supernatant after centrifugation, which is an inverse measure of phase separation, changes with salt concentration. hnRNPA2 LC shows low phase separation (high protein remaining in the supernatant) at low salt conditions and near-complete phase separation (no protein remaining in the supernatant) at high salt concentrations. In contrast, tyrosine phosphorylated hnRNPA2 LC shows higher phase separation (less protein remaining in the supernatant) at low salt concentrations than at high salt concentrations. Conditions: 20 µM protein, 20 mM MES pH 5.5 150 mM urea, salt concentration as indicated, 25°C. Error bars are standard deviation of three replicates.
B  While hnRNPA2 LC shows no droplets in low salt conditions and droplets in high salt conditions, tyrosine phosphorylated hnRNPA2 LC shows droplets in all salt concentrations tested, although more droplets are present at low salt conditions. Conditions: 20 µM proteins, 20 mM MES pH 5.5 150 mM urea with 0 mM, 50 mM, or 500 mM NaCl as indicated. Scale bar: 10 µm.
C  Fluorescence micrographs showing that although TOG D1 and hnRNPF PLD partition into hnRNPA2 LC droplets (rows 1 and 3), they are unable to partition into tyrosine phosphorylated hnRNPA2 LC droplets (rows 2 and 4). Conditions: 20 µM proteins (~1% fluorescently labeled), 20 mM MES pH 5.5, 50 mM NaCl, 150 mM urea. Scale bar: 10 µm.
D  Phosphomimic variants containing 5 or 12 serine to glutamate substitutions (hnRNPA2 LC$^{5E}$ and hnRNPA2 LC$^{12E}$, respectively) both allow partitioning of TOG D1 and hnRNPF PLD, even though hnRNPA2 LC$^{12E}$ barely phase separates at these conditions, indicating that increased negative charge is insufficient to prevent partitioning of TOG D1 and hnRNPF PLD. See Appendix Fig S7B for quantification of LLPS of hnRNPA2 LC$^{5E}$ and hnRNPA2 LC$^{12E}$. Conditions: 20 µM proteins (~1% fluorescently labeled), 20 mM MES pH 5.5, 50 mM NaCl, 150 mM urea. Scale bar: 10 µm.

phosphorylation alters hnRNPA2 LC self-interactions. The disordered monomeric form of hnRNPA2 LC expands as the number of tyrosine phosphorylations increases (Fig 4C), consistent with decreased interactions between the many sites of interaction (Martin *et al*, 2020). Additionally, hnRNPA2 LC WT, D290V, and P298L all have substantially reduced probability of forming intramolecular contacts when phosphorylated (Fig 4D, Appendix Fig S9B and C). These computational assessments further support a model based on our *in vitro* studies where tyrosine phosphorylation reduces hnRNPA2 phase separation and aggregation.

### Stress alters HRPA-1 assembly in *C. elegans* neurons

We next moved to *C. elegans* to assess the impact of tyrosine phosphorylation on hnRNPA2 self-association and toxicity *in vivo*. Although the RRMs of hnRNPA2 homologs are well conserved across animals including in *C. elegans* (55% sequence identity), the disordered domain is not well conserved at the sequence position level across animal species. Therefore, we examined the conservation of hnRNPA2 LC sequence characteristics between humans, vertebrates, and invertebrates. Human hnRNPA2 LC (Appendix Fig S10A) has an unusually high proportion of glycine (46% of residues) compared with the average intrinsically disordered region (IDR) or globular protein (Tompa, 2002) (Appendix Fig S10B and C). We found that the high glycine character was conserved both in vertebrates, with rats, mice, and *Xenopus* all having 46–47% glycine, and in invertebrates, where *Drosophila* Hrb87F and *C. elegans* HRPA-1, the orthologs of hnRNPA2, have LC domains with 52–53% glycine (Appendix Fig S10D–H). However, the *C. elegans* LC has a higher

reduction in prion-like sequence composition and zipper amyloid stability may explain both the altered phase separation and reduced aggregation of the tyrosine phosphorylated hnRNPA2. Finally, we modified an established coarse grain simulation method (Dignon *et al*, 2018) to incorporate pY residues to evaluate how

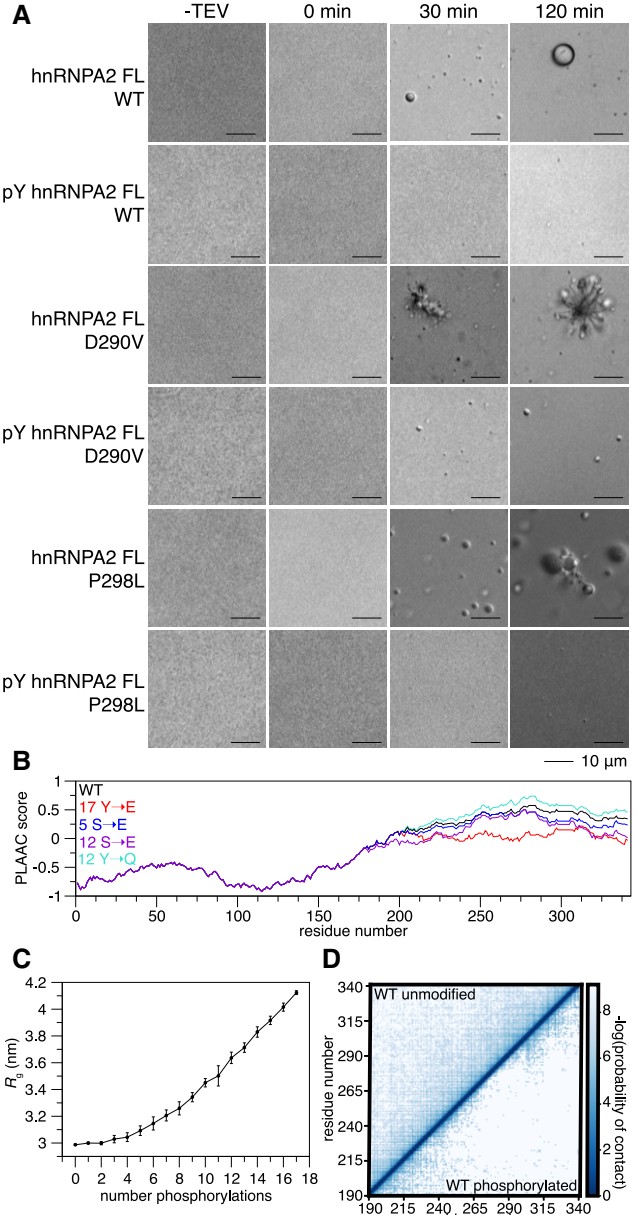

**A**

|  | -TEV | 0 min | 30 min | 120 min |
|---|---|---|---|---|
| hnRNPA2 FL WT | | | | |
| pY hnRNPA2 FL WT | | | | |
| hnRNPA2 FL D290V | | | | |
| pY hnRNPA2 FL D290V | | | | |
| hnRNPA2 FL P298L | | | | |
| pY hnRNPA2 FL P298L | | | | |

—— 10 µm

**B**

WT
17 Y→E
5 S→E
12 S→E
17 Y→Q

PLAAC score — residue number

**C**

$R_g$ (nm) — number phosphorylations

**D**

WT unmodified / WT phosphorylated — residue number
-log(probability of contact)

**Figure 4.** Tyrosine phosphorylation reduces contacts and aggregation of disease mutants. See also Appendix Fig S9.

A  After cleavage of a C-terminal maltose-binding protein solubility tag, hnRNPA2 FL WT undergoes LLPS and disease mutants D290V and P298L aggregate. In contrast, phosphorylated hnRNPA2 FL WT and P298L do not undergo LLPS or aggregation in the time frame tested. Phosphorylated hnRNPA2 FL D290V can form some structures resembling liquid droplets but does not form amorphous aggregates in the time frame tested. Conditions: 20 µM proteins, 20 mM Tris pH 7.4, 50 mM NaCl. Scale bar: 10 µm.

B  PLAAC analysis (Lancaster *et al*, 2014) of hnRNPA2 (black), changing all tyrosine residues in hnRNPA2 LC to glutamate (17 Y→E, red), 5 serine to glutamate phosphomimic mutations (5 S→E, blue), all serine to glutamate (12 S→E, purple), and all tyrosine to glutamine (17 Y→Q) shows that hnRNPA2 LC prion-like character reduces slightly with serine phosphomimic mutations, but decreases drastically with tyrosine phosphomimic mutations.

C  As the number of phosphorylation events increases, the radius of gyration ($R_g$) of hnRNPA2 LC increases in coarse-grained simulations. $R_g$ uncertainty bars represent S.E.M. from 10 different simulations with the same number of phosphorylated tyrosines randomly placed in different positions.

D  Coarse-grained simulations of WT show that compared to the unphosphorylated form, phosphotyrosine hnRNPA2 LC forms fewer contacts with itself.

proportion of glutamine and lower proportion of serine than the human LC. Interestingly, LC length is similar among species as well. Given this similarity in residue composition between humans and *C. elegans* in LC domains, we hypothesized that *C. elegans* HRPA-1 ability to self-assemble is conserved. To test this biochemically, we generated recombinant maltose-binding protein-tagged full-length HRPA-1 and found that cleavage of the MBP-tag induced HRPA-1 *in vitro* phase separation (Appendix Fig S10I), as observed previously for human hnRNPA2 FL (Ryan *et al*, 2018). Additionally, we generated recombinant HRPA-1 LC and found this domain was soluble in low salt conditions (0 mM), like hnRNPA2 LC. At 50 mM salt, HRPA-1 LC formed amorphous aggregates (Appendix Fig S10J), rather than liquid droplets like hnRNPA2 LC (Ryan *et al*, 2018), though both observations are signs of avid self-association. Given the similarity between these LC domains, we created *C. elegans* strains expressing

a chimeric HRPA-1 protein replacing most of the LC sequence with the corresponding human hnRNPA2 LC sequence.

To visualize self-association in *C. elegans*, we created transgenic animals expressing HRPA-1 tagged with the red fluorescent protein mScarlet (Bindels *et al*, 2017) inserted between the RRMs and LC (HRPA-1mScarlet) (Fig 5A). This construct was expressed in glutamatergic neurons using the *mec-4* promoter and *hrpa-1* 3′UTR (Fig 5B and C). To examine the effect of the D290V disease mutation on self-association, we also replaced the third coding exon of the *C. elegans hrpa-1* gene with the corresponding human (*Homo sapiens*, Hs) protein sequence codon optimized for *C. elegans* expression, with the human wild-type sequence (HRPA-1HsLC$^{WT}$mScarlet) or with the disease-causing D290V substitution (HRPA-1HsLC$^{D290V}$mScarlet). First, we tested whether the *C. elegans* HRPA-1mScarlet and chimeric HRPA-1HsLC$^{WT}$mScarlet had similar levels of self-association *in vivo* and if exposure to oxidative stress altered self-association in neuronal processes (Fig 5D). As it is unclear if HRPA-1 assemblies we observed are stress granules, transport granules, or aggregates, the observed assemblies are referred to herein as "spots". Without stress, few neuronal processes contained HRPA-1mScarlet or HRPA-1HsLC$^{WT}$mScarlet spots and there was no difference between transgenes or lines (Fig 5D). After exposure to 22 h of paraquat-induced oxidative stress, spots increased for both transgenes and all lines. HRPA-1mScarlet and HRPA-1HsLC$^{WT}$mScarlet were indistinguishable in their response to stress (Fig 5D). We concluded that oxidative stress similarly increases HRPA-1mScarlet and HRPA-1HsLC$^{WT}$mScarlet spot formation in these neuronal processes and undertook the next studies using only HRPA-1HsLC$^{WT}$mScarlet.

To test whether tyrosine phosphorylation alters self-association *in vivo*, we generated transgenes expressing a constitutively active variant of Fyn kinase (Y531F) (Hirose *et al*, 2003) under the same *mec-4* promoter (Fig 5E). We generated new lines carrying either HRPA-1HsLC$^{WT}$mScarlet or HRPA-1HsLC$^{D290V}$mScarlet along with either constitutively active Fyn (Fyn*) or an empty control with a frame shift mutation in the Fyn coding region (Fyn$^{empty}$). In the absence of stress, there was no significant difference in spots

observed between either HRPA-1HsLC^WTmScarlet or HRPA-1HsLC^D290VmScarlet with or without activated Fyn (Fig 5F). There was not a significant difference after exposure to 22 h of paraquat-induced oxidative stress either (Fig 5G). However, we noted a tendency repeated in both unstressed and stressed conditions for animals expressing both Fyn* and HRPA-1HsLC^D290VmScarlet to have fewer spots than animals expressing HRPA-1HsLC^D290VmScarlet and Fyn^empty control. Additionally, HRPA-1HsLC^D290VmScarlet showed significantly more spots than HRPA-1HsLC^WTmScarlet in the absence of Fyn expression (Fyn^empty) both with and without

stress, suggesting that the mutation alters the self-assembly of the chimera in these neurons. Of note, we did not observe any degeneration in these neurons so observed spots may not be a toxic species.

**Chimeric *hrpa-1*HsLC^D290V causes neurodegeneration in *C. elegans*, which is rescued by *tdp-1* deletion or expression of active tyrosine kinase Fyn**

Although expression of activated Fyn did not significantly change HRPA-1 self-association *in vivo* as monitored by spot formation, we

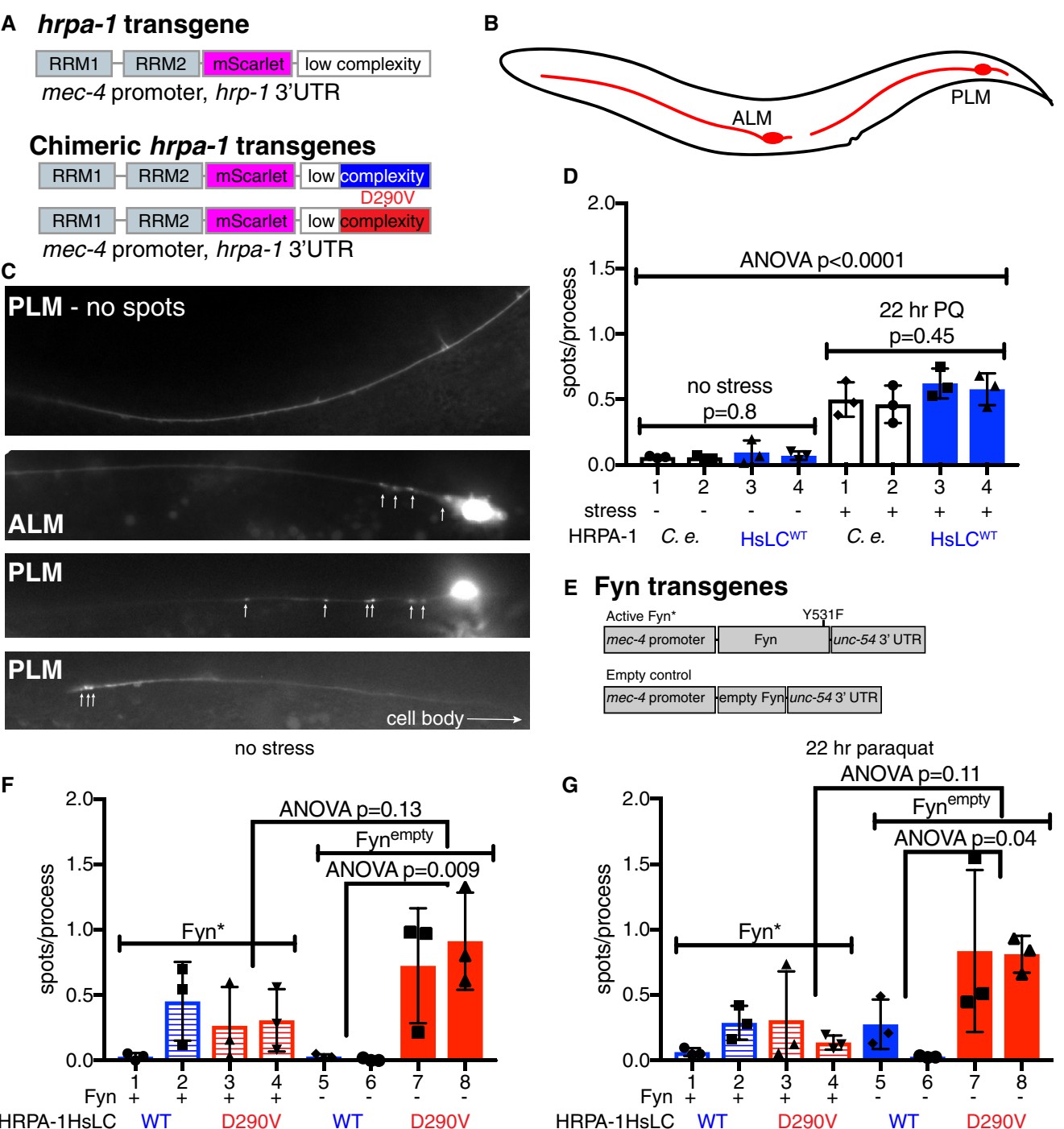

Figure 5.

◄

**Figure 5.   Stress alters HRPA-1 assembly in vivo. See also Appendix Figs S10 and S12 and Appendix Tables S1-S2.**

A   mScarlet was added between the RNA recognition motifs (RRMs) and LC of HRPA-1, and the third exon was replaced with the corresponding human sequence; disease mutation D290V was introduced.

B   Glutamatergic touch neurons ALM and PLM expressing *mec-4* were scored for number of spots in processes.

C   Representative images of *mec-4* neurons with and without HRPA-1mScarlet spots. (various HRPA-1mScarlet containing genotypes.)

D   After 22 h of paraquat-induced oxidative stress, both HRPA-1mScarlet and HRPA-1HsLC$^{WT}$mScarlet assemble more than when unstressed (ANOVA $P < 0.0001$). No significant differences were found within the no stress ($P = 0.8$) and stress groups ($P = 0.45$) by ANOVA, indicating that there is no difference in assembly between HRPA-1mScarlet and HRPA-1HsLC$^{WT}$mScarlet. $N = 12$ animals per genotype.

E   Fyn expression constructs including activated Fyn (Fyn*, Y531F) and empty control Fyn$^{empty}$.

F   There is no significant difference in number of spots in HRPA-1HsLC$^{WT}$mScarlet lines with or without Fyn, although there seems to be a trend of reduced spots in HRPA-1HsLC$^{D290V}$mScarlet lines in animals also expressing Fyn*; this difference is not statistically significant (ANOVA). $N = 12$ animals per genotype.

G   After exposure to 22 h of paraquat-induced oxidative stress, there is no significant difference between HRPA-1HsLC$^{WT}$mScarlet lines with or without Fyn, although there seems to be a trend of reduced spots in HRPA-1HsLC$^{D290V}$mScarlet lines in animals also expressing Fyn*; this difference is not statistically significant (ANOVA). $N = 12$ animals per genotype.

Data information: The mean with S.E.M. is reported. Three independent trials for all determinations were performed, with experimenter blinded to genotype for each trial.

hypothesized that even a slight decrease in D290V association might alter pathological outcomes. To test this, we examined the consequences of manipulating *hrpa-1* expression on neurodegeneration. We generated an *hrpa-1* genomic rescue construct as before, in which the third coding exon of the *C. elegans hrpa-1* gene was replaced with the corresponding human protein sequence (optimized for *C. elegans* expression) resulting in the expression of an un-tagged chimeric protein under control of the *hrpa-1* promoter and 3′UTRs (Fig 6A, Appendix Fig S11A). We generated additional transgenic lines expressing the disease variant (D290V), lines expressing a truncated form removing the LC domain with a stop codon at the beginning of the 3$^{rd}$ coding exon (ΔLC lines), and control lines with only the plasmid vector and no *hrpa-1* DNA (empty). No overt behavioral or morphological defects were observed in any lines. To assess neurodegeneration, we used dye uptake assays, which provide a sensitive readout of glutamatergic neurodegeneration in other *C. elegans* disease models (Faber *et al*, 1999; Baskoylu *et al*, 2018). In dye uptake assays, the intact sensory endings of specific glutamatergic sensory neurons take up fluorescent dyes from the environment and backfill cell bodies. Either cell death or process degeneration prevents dye uptake, providing an assessment of neuron integrity. However, even after 22 h of paraquat-induced oxidative stress, no animals from any line tested had glutamatergic neurodegeneration as assayed by dye update (Appendix Fig S11B) although the WT and D290V chimeras were expressed (Appendix Fig S12).

Realizing that a functional endogenous *C. elegans hrpa-1* gene might mitigate the consequences of chimeric HRPA-1 transgenes, we crossed the chimeric transgenes onto two available *hrpa-1(Δ)* loss-of-function alleles (*tm781* and *ok592*). Knockdown or loss of *hrpa-1* causes sterility and partially penetrant lethality (Appendix Fig S11C) (Longman *et al*, 2000; Rual *et al*, 2004; Fernandez *et al*, 2005; Sonnichsen *et al*, 2005); therefore, it was important to confirm that the chimeric transgenes rescue this phenotype. Additionally, all lines carrying *hrpa-1(Δ)* had to be maintained over a genetic balancer that was evicted for assays, so lines were compared with the normal lab strain (N2, "WT") as well as to WT animals from the same genetic background (i.e., from mothers carrying one copy of the balancer, "+"). Indeed, about half of the homozygous *hrpa-1(Δ)* animals failed to lay eggs by the fourth day of adulthood (Appendix Fig S11C), but the expression of *hrpa-1*HsLC$^{WT}$ restored egg laying (Appendix Fig S11C). *hrpa-1(Δ)* sterility was also rescued by most of the *hrpa-1*HsLC$^{D290V}$ lines (Appendix Fig S11C). Neither

*hrpa-1*HsLC$^{empty}$ or *hrpa-1*HsLC$^{ΔLC}$ rescued sterility (Appendix Fig S11C), indicating that phenotypic rescue requires an intact LC domain, not simply the RNA-binding regions of HRPA-1. Combined, these data show that the glycine-rich LC domain of *C. elegans* HRPA-1 is important for function and that the glycine-rich LC domain from human hnRNPA2 can substitute *in vivo* for the orthologous *C. elegans* HRPA-1 LC domain, indicating functional conservation.

Next, we examined glutamatergic neuron degeneration on the *hrpa-1* loss-of-function background to determine whether *hrpa-1*HsLC$^{D290V}$ causes neurodegeneration. Even in the absence of stress, *hrpa-1* loss-of-function resulted in partially penetrant dye uptake defects (Appendix Fig S11D). Roughly half the tail or head neurons failed to uptake dye (Appendix Fig S11D) but the neuron cell bodies were present when scored for the expression of *osm-10*p::GFP (Appendix Fig S6E), indicating that the neurons are degenerating but not dying. This dye uptake defect was rescued by introduction of the chimeric *hrpa-1*HsLC$^{WT}$ or *hrpa-1*HsLC$^{D290V}$ transgenes, but not the empty control *hrpa-1*HsLC$^{empty}$ (Fig S6E). Therefore, *hrpa-1* is important for neuron survival and D290V does not dramatically impair chimeric HRPA-1 function without stress. However, after 22 h of paraquat-induced oxidative stress, *hrpa-1*HsLC$^{D290V}$ animals had increased glutamatergic neurodegeneration in their phasmid neurons (but not amphid neurons), while *hrpa-1*HsLC$^{WT}$ animals remained unaffected (Fig 6B, Appendix Fig S11F). Neurons are degenerated, not dead, as all cell bodies were accounted for when neurons were scored for the expression of *osm-10*p::GFP after exposure to stress (Appendix Fig S11G). We conclude that application of oxidative stress reveals neurodegeneration caused by the deleterious impact of the disease-associated D290V mutation on chimeric HRPA-1 function.

To reduce the number of alleles tested going forward, we decided to perform further assays on a single *hrpa-1(Δ)* deletion allele. In both fertility and glutamatergic neurodegeneration assays, *hrpa-1 (ok592)* was less severe than *hrpa-1(tm781)*, both alone and with *hrpa-1*HsLC$^{D290V}$. The *ok592* deletion removes sequences encoding the end of the second RRM domain and the entire LC domain; the first RRM is completely intact. In contrast, *hrpa-1(tm781)* is likely a complete loss of function (null), as the deletion removes RRM1 coding sequences and frame shifts subsequent translation. Therefore, we used *hrpa-1(tm781)* for the remaining studies. As cholinergic motor neuron loss is a feature of MSP in addition to glutamatergic neuron loss (reviewed in Benatar *et al* (2013)), we

sought to determine the impact of D290V on *C. elegans* motor neuron loss, using a cholinergic neuron-specific GFP transgene to visualize these cells. However, both without stress and after 22 h of paraquat stress, none of the transgenic lines showed cholinergic motor neuron loss compared with control lines (WT or +) (Appendix Fig S11H), suggesting that our model does not recapitulate specific cholinergic neuron defects associated with human motor neuron disease.

Previous studies suggest that the D290V may cause gain-of-function defects (reviewed in Kapeli *et al* (2017)) as RNA splicing

changes caused by the mutation are not equivalent to loss of hnRNPA2 (Martinez *et al*, 2016) and D290V causes aggregation and localization to stress granules (Kim *et al*, 2013; Martinez *et al*, 2016) as well as enhanced cell death and stress response in motor neuron culture (Martinez *et al*, 2016). Yet, our observations above show stress-induced degeneration for both the chimeric *hrpa-1*HsLC$^{D290V}$ and the *hrpa-1*($\Delta$) deletion allele, suggesting that D290V-associated degeneration in our model may result from either loss or gain of function. Here, we were afforded the opportunity to evaluate whether gain-of-function mechanisms contribute to the D290V-

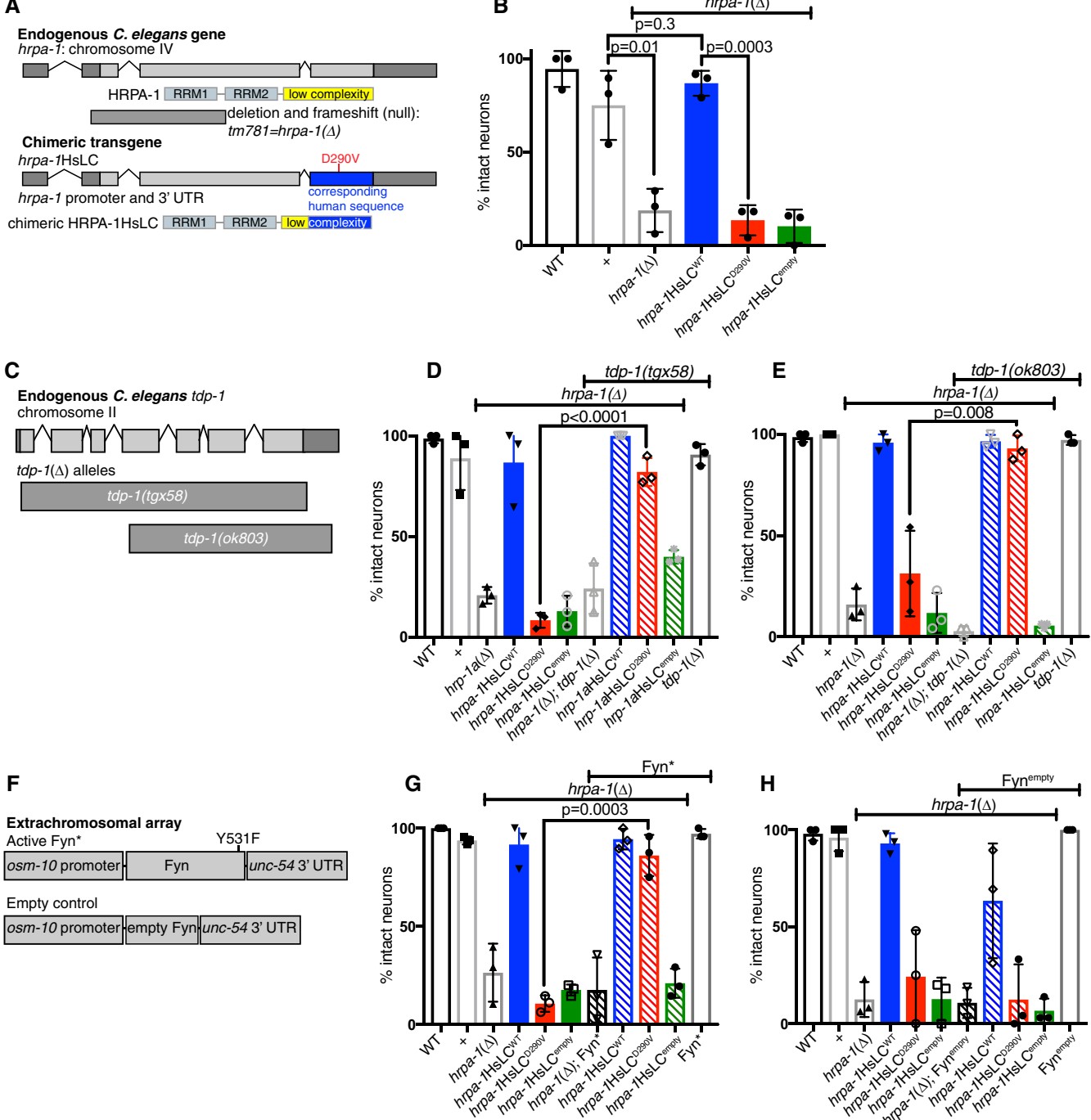

**Figure 6.**

◀

**Figure 6.  hnRNPA2-D290V causes *C. elegans* neurodegeneration, which is rescued by *tdp-1* deletion or overexpression of active Fyn kinase. See also Appendix Figs S11 and S12, and Appendix Tables S1–S2.**

A    Top: schematic depicting *hrpa-1* gene and a loss-of-function deletion allele (*tm781*), referred to as *hrpa-1(Δ)*. Bottom: To create a chimeric protein (HRPA-1HsLC), the third coding exon (blue) was replaced with a codon-optimized sequence encoding the majority of the human LC.

B    After 22 h of paraquat-induced oxidative stress, *hrpa-1(Δ)* animals showed increased degeneration of tail glutamatergic (phasmid) neurons, based on defective dye uptake. Defective dye uptake indicates either neuron process degeneration or death. This defect is rescued by introduction of *hrpa-1*HsLC$^{WT}$, but not *hrpa-1*HsLC$^{D290V}$ or *hrpa-1*HsLC$^{empty}$. N = 6–12 animals/genotype/trial, significance from two-tailed t-test.

C    The *C. elegans* ortholog of TDP-43, *tdp-1*, contains 7 exons. A new deletion allele, *tgx58*, deletes the entire coding region of *tdp-1*. A second deletion allele, *ok803*, removes part of the 4$^{th}$ exon and the remaining three exons.

D    Loss of *C. elegans* TDP-43 ortholog, *tdp-1*, rescues glutamatergic tail/phasmid neurodegeneration caused by *hrpa-1*HsLC$^{D290V}$ after 22 h of paraquat-induced oxidative stress, but not *hrpa-1(Δ)*. N = 4–12 animals/genotype/trial, significance from two-tailed t-test; *tdp-1Δ*=*tdp-1(tgx58)*.

E    Loss of *C. elegans* TDP-43 ortholog, *tdp-1*, rescues glutamatergic tail/phasmid neurodegeneration caused by *hrpa-1*HsLC$^{D290V}$ after 22 h of paraquat-induced oxidative stress, but not *hrpa-1(Δ)*. N = 4–12 animals/genotype/trial, significance from two-tailed t-test; *tdp-1Δ*=*tdp-1(ok803)*.

F    A constitutively activated Y531F Fyn (Fyn*) construct was expressed in the glutamatergic sensory tail (phasmid) neurons using the *osm-10* promoter. An empty construct removing the beginning of Fyn and causing a frame shift mutation was constructed similarly.

G    The expression of Fyn* in sensory glutamatergic neurons (*osm-10* promoter, *unc-54* 3′UTR) reduces glutamatergic tail/phasmid neuron degeneration in *hrpa-1*HsLC$^{D290V}$ animals after 22 h of paraquat-induced oxidative stress. N = 9–12 animals/genotype/trial, significance from two-tailed t-test.

H    The expression of an empty Fyn control (Fyn$^{empty}$) in sensory glutamatergic neurons (*osm-10* promoter, *unc-54* 3′UTR) does not alter glutamatergic phasmid neuron degeneration in *hrpa-1*HsLC$^{D290V}$ animals after 22 h of paraquat-induced oxidative stress. N = 9–12 animals/genotype/trial.

Data information: The mean with SEM is reported. Three independent trials for all determinations were performed, with experimenter blinded to genotype for each trial. In panels (B, D, E, G, H), *hrpa-1*HsLC$^{WT}$, *hrpa-1*HsLC$^{D290V}$, and *hrpa-1*HsLC$^{empty}$ were maintained on a *hrpa-1(Δ)/tmC25[tmIs1241]* background, but transgenes were tested in animals homozygous for *hrpa-1(Δ)*. WT is N2, the standard lab strain, while "+" is the N2 *hrpa-1* allele maintained over *tmC25[tmIs1241]* and assayed as a homozygote.

associated degeneration we observe. We focused on the ALS/FTD-associated RNA-binding protein TDP-43 because hnRNPA2 and TDP-43 are thought to physically interact to mediate RNA processing (Buratti *et al*, 2005; D'Ambrogio *et al*, 2009) and they co-aggregate in patients (Kim *et al*, 2013) and *in vitro* (Ryan *et al*, 2018). Based on these potential connections, we then sought to interrogate a possible link between TDP-1, the ortholog of TDP-43, and mutant hnRNPA2-associated neurodegeneration in our *C. elegans* model. Given that we are not currently able to visualize protein interactions or aggregation in the glutamatergic neurons in which we see degeneration, we decided to take a genetic model approach. We hypothesized that loss of the *tdp-1* gene might reduce glutamatergic neurodegeneration. Indeed, *tdp-1* loss of function (Fig 6C) did dramatically reduce glutamatergic neurodegeneration in the *hrpa-1*HsLC$^{D290V}$ animals but had little or no rescue in *hrpa-1(Δ)* animals (Fig 6D and E). Two different *tdp-1* loss-of-function alleles (*tgx58* or *ok803*) (Fig 6C) had the same beneficial impact (Fig 6D and E). As such, *tdp-1* expression is required for D290V-associated glutamatergic neurodegeneration in our *C. elegans* model. We note that our studies do not probe the mechanism linking *tdp-1* to neurodegeneration and are limited to the native *C. elegans tdp-1*, which is not identical to TDP-43. Hence, the relevance of these findings to TDP-43 function in mammalian cells and dysfunction in human disease requires further elucidation. Although the mechanistic connection between *tdp-1* and hnRNPA2 mutant-associated neurodegeneration remains to be explored further, *tdp-1* loss-of-function alleles provide critical insight into this model; *hrpa-1*HsLC$^{D290V}$ does not cause a strictly loss-of-function defect, distinguishing the D290V-associated defect from neurodegeneration associated with *hrpa-1* deletion.

As *hrpa-1*HsLC$^{D290V}$ animals show neurodegeneration with gain-of-function attributes that may be related to cytoplasmic or nuclear aggregation of hnRNPA2 (reviewed in Kapeli *et al* (2017)), we hypothesized that preventing aberrant self-interaction of the chimeric hnRNPA2 D290V may prevent toxicity. Given that phosphorylation of hnRNPA2 LC prevents aggregation (see above), we tested the effect of expression of constitutively active form of Fyn kinase (Fyn*), which is known to phosphorylate hnRNPA2 (White

*et al*, 2008), on hnRNPA2-associated neurodegeneration. To do so, we expressed Fyn* in the glutamatergic neurons affected in *hrpa-1*HsLC$^{D290V}$ animals (Fig 6F) and examined the consequences of Fyn* expression on stress-induced neurodegeneration. We found animals expressing both *hrpa-1*HsLC$^{D290V}$ and Fyn* had reduced stress-induced neurodegeneration when compared to *hrpa-1*HsLC$^{D290V}$ animals lacking Fyn expression (Fig 6G). In contrast, the expression of an empty control (with most of the coding region of Fyn removed, Fyn$^{empty}$) did not alter neurodegeneration in *hrpa-1*HsLC$^{D290V}$ animals (Fig 6H). To confirm these results, we generated independent transgenes of both Fyn* and Fyn$^{empty}$ and found the same result: Active Fyn reduced neurodegeneration in *hrpa-1*HsLC$^{D290V}$ animals while Fyn$^{empty}$ did not (Appendix Fig S11I). No rescue was observed for the defects in *hrpa-1* deletion alone, suggesting that the Fyn*-dependent rescue is specific for D290V-associated neurodegeneration and again distinguishing D290V from *hrpa-1* deletion-associated defects. We note that, given the small number of cells in which Fyn is expressed and hence an extremely low abundance of protein, we were not able to confirm that hnRNPA2 phosphorylation is enhanced in the chimeric hnRNPA2 D290V in these cells. However, as active Fyn was expressed only in the neurons that degenerate we also can conclude that Fyn has a cell autonomous impact on *hrpa-1*HsLC$^{D290V}$-associated neurodegeneration.

## Discussion

Although RNA granules are heterogeneous mixtures of proteins and RNAs, much work on LLPS of disease-associated proteins has focused on single proteins (Burke *et al*, 2015; Molliex *et al*, 2015; Patel *et al*, 2015; Conicella *et al*, 2016; Wang *et al*, 2018a). Here, we examine the distinct interactions between hnRNPA2 and two other hnRNPA2 transport granule components, hnRNPF and TOG (Fig 7). hnRNPF is readily incorporated into droplets formed by the low complexity domain of hnRNPA2 but not of FUS, suggesting that some specificity of pairwise interaction is encoded in distinct low

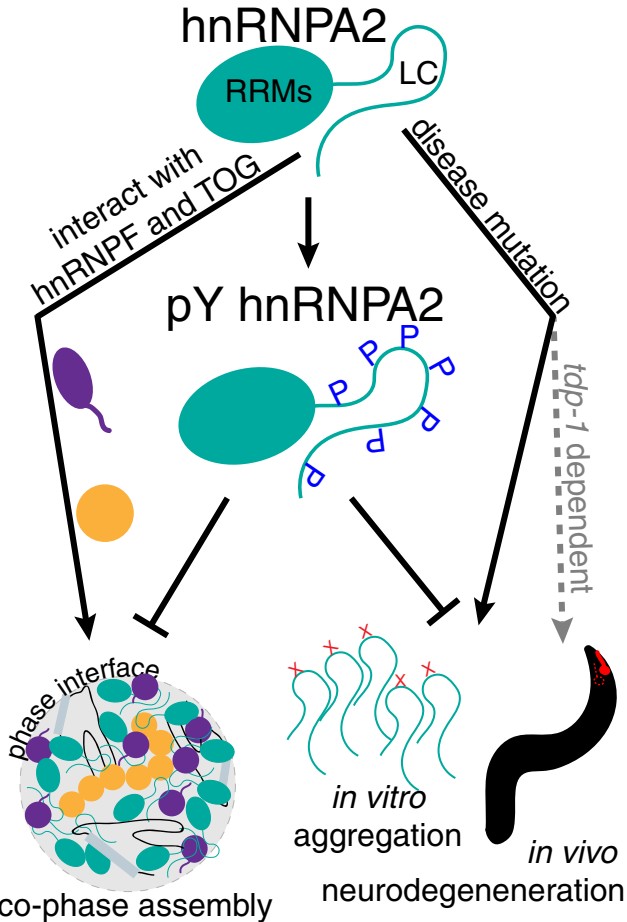

**Figure 7. hnRNPA2 interactions and neurodegeneration are altered by tyrosine phosphorylation.**

hnRNPA2 interacts with transport granule components hnRNPF and ch-TOG in the phase-separated state. Disease mutations (X) D290V and P298L induce aggregation of hnRNPA2 *in vitro*, while D290V induces *tdp-1*-dependent neurodegeneration in a *C. elegans* model. hnRNPA2 LC tyrosine phosphorylation alters hnRNPA2 LC LLPS *in vitro*, prevents interaction with hnRNPF and ch-TOG, reduces aggregation *in vitro*, and expression of an activated tyrosine kinase reduces D290V-associated neurodegeneration in the *C. elegans* model.

complexity domains. This specificity depends on the arginine residues in hnRNPA2, consistent with the important role for arginine in phase separation due to its unique modes of interaction (Chong *et al*, 2018; Vernon *et al*, 2018). In contrast, TOG, an α-helix rich globular protein, interacts with hnRNPA2 LC differently than other previously described globular TOG-binding partners (Slep & Vale, 2007; Ayaz *et al*, 2012). These other binding partners, including tubulin, primarily interact with TOG disordered loops (Ayaz *et al*, 2012) while hnRNPA2 LC weakly binds both the disordered loops and helical faces of TOG D1, which therefore may still permit simultaneous interaction with tubulin, possibly as part of granule transport. Of note, the binding affinity for TOG D1 and hnRNPA2 LC is very weak, in contrast to previous reports (Falkenberg *et al*, 2017). As the previous values were measured with full-length hnRNPA2, which we have shown is highly aggregation prone without a

solubilizing fusion, it is possible that these measurements were confounded by self-association and LLPS of hnRNPA2, rather than only a stoichiometric heterotypic interaction. Together, these findings show how protein–protein interactions in phase-separated assemblies could give rise to specificity of granule protein interactions and partitioning despite weak affinities. Conversely, our results also demonstrate that the toxicity caused by mutant hnRNPA2 expression, but not toxicity caused by loss of the *C. elegans* ortholog of hnRNPA2, is ameliorated by deletion of *tdp-1* (Fig 7), the ortholog of the RNA-binding protein TDP-43 that forms aggregates in the majority of ALS cases (Mackenzie & Rademakers, 2008) and co-aggregates with hnRNPA2 in MSP (Kim *et al*, 2013). Therefore, future work further elucidating how these proteins interact genetically, physiologically, and pathologically will be important for evaluating their combined contribution to neurodegeneration.

Our data suggest that the impact of phosphorylation on LLPS may be type and context specific. Serine/threonine phosphorylation in the low complexity domain of FUS reduces aggregation of FUS *in vitro* and *in vivo* (Monahan *et al*, 2017). Yet, serine/threonine phosphorylation increases LLPS of FMRP (Tsang *et al*, 2019) and is required for the interaction between FRMP and CAPRIN1 (Kim *et al*, 2019). We found that although both tyrosine phosphorylation and incorporation of negatively charged amino acids as phosphomimetics at serine positions alter LLPS of hnRNPA2 LC, serine phosphomimics do not prevent partitioning of hnRNPF and TOG, while tyrosine phosphorylation does prevent partitioning. This distinction in the ability of related chemical changes to elicit different specific effects suggests that serine and tyrosine phosphorylation may be used to differentially tune interactions between hnRNPA2 and other proteins. We note that hnRNPF is also known to be tyrosine phosphorylated by Fyn (White *et al*, 2012) and would expect that tyrosine phosphorylation of hnRNPF would likely contribute to granule dissolution by preventing interaction with hnRNPA2. Further, like in many granule proteins, hnRNPA2 arginine residues are asymmetrically dimethylated, which reduces LLPS (Ryan *et al*, 2018) and may also alter partitioning of transport granule components. Hence, phase separating proteins may encode specificity of partitioning not only by amino acid sequence (see above), but also by post-translational modification of specific residues, significantly expanding the effective amino acid sequence code and providing the ability to dynamically regulate interactions. Of note, tyrosine phosphorylation of ZBP1/IGF2BP1 also leads to local translation of mRNAs carried by this protein (Huttelmaier *et al*, 2005), suggesting tyrosine phosphorylation may be a common mechanism for initiating translation of transported mRNAs carried in transport granules.

Inhibition of tyrosine kinase pathways and Fyn kinase itself (Nygaard *et al*, 2014; Kaufman *et al*, 2015; Imamura *et al*, 2017; Nygaard, 2018; Smith *et al*, 2018) is potential therapeutic targets in Alzheimer's disease. Our data indicate that while tyrosine phosphorylation can reduce aggregation and phase separation *in vitro*, this does not clearly translate to substantial reduction of puncta formation in an *in vivo* model. However, due to experimental limitations of *C. elegans*, we assayed degeneration and puncta formation in distinct neurons. We also note that the toxic species is not well understood and may not be aggregates or may be smaller than the aggregates probed by fluorescence microscopy (Siddiqi *et al*, 2019). Yet, though expression of activated Fyn showed only a modest reduction in self-assembly *in vivo*, it led to a large reduction in

neurodegeneration in glutamatergic neurons in an animal model. As such, it is possible that aberrant assembly or aggregation of hnRNPA2 driven by disease mutation is toxic and the presence of these assemblies directly correlates with neurodegeneration, as has been shown for muscle degeneration (Kim *et al*, 2013). Furthermore, Fyn expression has no effect on degeneration associated with loss of *hrpa-1*, suggesting Fyn-associated rescue is related directly to mutant hnRNPA2 expression. In contrast, it is important to note that activated Fyn may have beneficial functions beyond reducing hnRNPA2 aggregation *in vivo* or may have an indirect effect on hnRNPA2 D290V-associated toxicity in our model. Hence, elucidating the impact of Fyn on mutant-associated defects in human neurons essential to evaluate the potential therapeutic benefit. As Fyn is a potential therapeutic target in other neurodegenerative diseases, these other mechanisms may be applicable to other causes of neurodegeneration beyond hnRNPA2. However, as reducing Fyn seems to be beneficial in other diseases (Nygaard *et al*, 2014; Kaufman *et al*, 2015; Imamura *et al*, 2017; Nygaard, 2018; Smith *et al*, 2018), it is possible that the impact of Fyn on neurodegeneration may be specific to the disease and expression may need to be modulated for subsets of neurons for a beneficial effect. These results from atomistic to animal models point to a potential for atomic details of interactions underlying normal granule assembly/disassembly to guide novel strategies to disrupt pathological dysfunction in neurodegenerative disease.

# Materials and Methods

## Experimental model and subject details

### C. elegans maintenance

*Caenorhabditis elegans* were maintained at 20°C or 25°C (for *mec-4*p::*hrpa-1*mScarlet extrachromosomal arrays only) under normal growth conditions. Experimenter was blinded to genotype for all trials. All trials were independent biological replicates. See Appendix Table S1 for a list of strains. In all figures except Appendix Fig S11B, *hrpa-1*HsLC^{WT}, *hrpa-1*HsLC^{D290V}, and *hrpa-1*HsLC^{empty} were maintained on a *hrpa-1(Δ)/tmC25[tmIs1241]* background, but transgenes were tested in animals homozygous for *hrpa-1(Δ)*. *hrpa-1(Δ)* was also always maintained over *tmC25[tmIs1241]* but homozygous *hrpa-1(Δ)* animals were assayed. In all figures, WT is N2, the standard lab strain, while "+" is the N2 *hrpa-1* allele maintained over *tmC25[tmIs1241]* and assayed as a homozygote.

### Bacterial culture

Uniformly $^{15}$N, $^{13}$C, or $^{2}$H labeled proteins were expressed in M9 in $H_2O$ or $^{2}H_2O$ with $^{15}$N ammonium chloride as the sole nitrogen source or $^{13}$C-glucose or $^{2}$H $^{13}$C-glucose as the sole carbon source as appropriate. Unlabeled proteins were expressed in LB. Cell pellets were harvested from 1 l cultures induced with IPTG at an OD600 of 0.6-1 after 4 h at 37°C. Tyrosine phosphorylated hnRNPA2 was grown in TKB1 competent cells (Agilent, 200134). 1 L cultures were grown at 18°C overnight to an OD600 of about 1, when there were induced with IPTG for 3 h at 37°C. Cells were then spun down at 2000 *g* for 10 min and exchanged into 1× TK media made fresh according to manufacturer's instructions. Cells were induced in TK media for 2 h at 37°C before harvesting the cell pellet.

Phosphorylated protein yield was low after full purification, so typically 6-12 l of each construct was grown simultaneously to get sufficient yield to freeze purified protein at greater than 1.1 mM. hnRNPA2 LC, FUS LC, hnRNPF PLD, and TOG D1 cell pellets were resuspended in 20 mM NaPi pH 7.4, 300 mM NaCl, 10 mM imidazole, 1 mM DTT while MBP-tagged full-length protein pellets were resuspended in 20 mM NaPi pH 7.4, 1 M NaCl, 10 mM imidazole, 1 mM with a Roche Complete EDTA-free protease inhibitor. Resuspended pellets were lysed on an Emulsiflex C3 and the cell lysate cleared by centrifugation (20,000 *g* for 60 min at 4°C).

## Method details

### Recombinant protein
#### Constructs
The following constructs and general purification strategies were used for protein expression in BL21 Star (DE3) *E. coli* cultures (Life Technologies):

- hnRNPA2 LC (190–341), insoluble His-tag purification as described (Ryan *et al*, 2018) (Addgene ID: 98657)
- hnRNPA2 LC S285C and S329C variants for PRE, insoluble His-tag purification as described (Ryan *et al*, 2018) (Addgene ID: 98665, 98667, respectively)
- MBP-hnRNPA2 LC, soluble His-tag purification as described (Ryan *et al*, 2018) (Addgene ID: 98661)
- FUS LC and FUS LC 12E, soluble His-tag purifications as described (Monahan *et al*, 2017) (Addgene ID: 98653, 98654)
- C-terminal maltose-binding protein-tagged hnRNPA2 FL WT, D290V, and P298L, soluble His-tag purification (Addgene ID: 139109, 139110, 139111, respectively)
- TOG D1, soluble His-tag purification (Addgene ID: 139112)
- hnRNPF PLD, His-tag purification (Addgene ID: 139113)
- N-terminal maltose-binding protein-tagged hnRNPF FL and hnRNPF ΔPLD, soluble His-tag purification (Addgene ID: 139114, 139115, respectively)
- HRPA-1 LC, insoluble His-tag purification (Addgene ID: 139116)
- C-terminal maltose-binding protein-tagged HRPA-1 FL, soluble His-tag purification (Addgene ID: 139117)
- hnRNPA2 LC$^{CD}$, hnRNPA2 LC$^{CD,R}$, hnRNPA2 LC$^{R \to K}$, hnRNPA2$^{N \to S}$, hnRNPA2 LC$^{5E}$, hnRNPA2 LC$^{12E}$, insoluble His-tag purification (Addgene ID: 139118, 139119, 139120, 139121, 139122, 139123, respectively)
- hnRNPF PLD$^{Y \to S}$, hnRNPF PLD$^{S \to A}$, His-tag purification (Addgene ID: 139124, 139125, respectively)
- FUS LC$^{CE}$, FUS LC$^{CE,R \to K}$, FUS LC$^{R}$, insoluble His-tag purification (Addgene ID: 139126, 139127, 139128)

### Protein purification

hnRNPA2 LC constructs were purified as described (Ryan *et al*, 2018). FUS constructs were purified as described (Monahan *et al*, 2017). MTSL-labeled hnRNPA2 LC constructs were purified and labeled as described (Ryan *et al*, 2018). Samples for NMR spectroscopy were produced in M9 minimal media with $^{2}$H, $^{15}$N, and $^{13}$C precursors as appropriate for the experiment.

Briefly, hnRNPA2 LC, variants, and HRPA-1 LC were expressed in *E. coli*. Inclusion bodies were resuspended in 8 M urea, 20 mM NaPi pH 7.4, 300 mM NaCl, 10 mM imidazole, and cleared by

centrifugation at 47,850 *g* for 60 min at 4°C. The cleared supernatant was filtered using a 0.2-µm filter and loaded onto a HisTrap 5 ml column (GE). Protein was eluted in a gradient of 10–300 mM imidazole over five column volumes. Fractions containing hnRNPA2 LC were pooled, concentrated, and diluted into pH 5.5 MES to a final urea concentration of less than 1 M. (pH 5.5 is chosen for optimal $^1$H $^{15}$N NMR signal due to $^1$H$_N$ water exchange rates. Biophysical behavior is similar at neutral pH and net charge of the protein is not expected to change significantly.) Protein was incubated with TEV protease at room temperature overnight. After TEV cleavage, protein was solubilized in 8 M urea and loaded on a HisTrap 5 ml column. Flow-through containing cleaved hnRNPA2 LC was collected, concentrated, buffer exchanged into 8 M urea, pH 5.5 MES (pH adjusted with Bis–Tris). Protein was flash-frozen at concentrations greater than 1.1 mM.

hnRNPF PLD and variants were purified from *E. coli* lysate by loading cleared soluble lysate onto a HisTrap 5 ml column after filtration by a 0.2-µm filter and then adding the insoluble fraction of the lysate after resuspension in 8 M urea buffer, clearing lysate by centrifugation at 47,850 *g* for 60 min at 4°C, and filtering with a 0.2-µm filter. The mixed soluble and insoluble protein was then eluted in a gradient of 10 to 300 mM imidazole (in urea) over 5 column volumes. Collected protein was concentrated and diluted into 20 mM NaPi pH 7.4, 300 mM NaCl to a final imidazole concentration of < 50 mM. Protein was incubated with TEV overnight at room temperature. Cleaved protein was filtered then loaded onto a 5 ml HisTrap, and flow-through containing cleaved protein was collected, concentrated, and buffer exchanged into 20 mM MES pH 5.5 (pH adjusted with Bis–Tris). Protein was flash-frozen at concentrations less than 300 µM.

Cleared TOG D1 lysate was loaded onto a HisTrap 5 ml column after filtration. Protein was eluted in a gradient of 10–300 mM imidazole over 5 column volumes. Fractions containing TOG D1 were pooled and loaded onto a Superdex 75 sizing column equilibrated in 20 mM NaPi 300 mM NaCl 1 mM DTT. Fractions containing TOG D1 without contaminants were pooled and cleaved by TEV at room temperature overnight. Cleaved protein was filtered and loaded onto a HisTrap 5 ml column and flow-through collected, concentrated, and buffer exchanged into 20 mM MES pH 5.5 (pH adjusted in Bis–Tris), and flash-frozen at concentrations greater than 1 mM.

Phosphorylated hnRNPA2 LC (and control not phosphorylated protein) was purified from MBP-hnRNPA2 LC. Cleared MBP-hnRNPA2 LC lysate was filtered and loaded onto a HisTrap 5 ml column. Protein was eluted in a gradient of 10 to 300 mM imidazole over five column volumes. Fractions containing MBP-hnRNPA2 LC were pooled and loaded onto a Superdex 200 equilibrated in 20 mM NaPi pH 7.4 300 mM NaCl. Fractions containing MBP-hnRNPA2 LC with minimal degradation were pooled and cleaved with TEV overnight. Protein was solubilized with 8 M urea, filtered, and loaded onto a HisTrap 5 ml. Flow-through containing cleaved hnRNPA2 LC was collected, concentrated, and buffer exchanged into 8 M urea 20 mM Bis–Tris pH 5.8. Protein was loaded onto a monoQ column equilibrated in the same buffer, and phosphorylated protein was eluted with a gradient to 1 M NaCl over 7 column volumes. Unphosphorylated protein was collected from the flow-through (expressed in BL21 cells due to leaky expression of the tyrosine kinase in TKB1 cells resulting in phosphorylation of hnRNPA2 even in the absence of kinase induction). Collected protein was concentrated, buffer exchanged into 8 M urea pH 5.5 MES (pH adjusted with Bis–Tris), and flash-frozen at concentrations greater than 1.1 mM.

Cleared maltose-binding protein-tagged full-length protein lysate was filtered and loaded onto a HisTrap 5 ml column. Protein was eluted in a gradient of 10–300 mM imidazole over 5 column volumes. Fractions containing MBP-FL were pooled and loaded onto a Superdex 200 sizing column equilibrated in 20 mM NaPi 1 M NaCl pH 7.4. Fractions containing undegraded protein were pooled, concentrated, and flash-frozen. Phosphorylated full-length hnRNPA2 was further subjected to a monoQ column equilibrated in 20 mM Tris pH 7.5 and eluted with a gradient to 1 M NaCl over 10 column volumes. Phosphorylated protein was collected, concentrated, buffer exchanged into 20 mM NaPi pH 7.4 1 M NaCl, and flash-frozen.

### AlexaFluor labeling

Proteins were labeled with NHS-ester AlexaFluors by diluting protein stocks into 20 mM NaPi pH 7.4 300 mM NaCl (1 M NaCl for FL hnRNPA2 and hnRNPF) with 8 M urea for insoluble proteins. AlexaFluor dissolved in DMSO was added at less than 10% total volume. Reactions were incubated for an hour, and then, free AlexaFluor was removed by desalting with 1 ml Zeba spin desalting columns equilibrated in the appropriate buffer for solubility. Labeled proteins were then concentrated and buffer exchanged into appropriate storage buffers and flash-frozen.

### Phase separation and microscopy

hnRNPA2 LC phase separation assays were performed as described (Ryan *et al*, 2018). Briefly, protein was diluted from 8 M urea into 20 mM MES pH 5.5 containing the appropriate salt concentration to a final protein concentration of 20 µM and final urea concentration of 150 mM. Samples were left to incubate at room temperature for 10 min and then spun down at 17,000 *g* for 10 min at room temperature. Protein concentration in the supernatant was measured by NanoDrop and calculated using the extinction coefficient of 25,330 M$^{-1}$ cm$^{-1}$.

Samples were prepared for microscopy by diluting proteins in appropriate conditions to final protein and urea (if applicable) concentrations. DIC images were taken with an Axiovert 200M microscope (Zeiss). Fluorescence microscopy images were taken on an LSM 710 or 880 (Zeiss). AlexaFluor-tagged proteins were doped in at 0.2 µl (< 1 µM final concentration) to prevent oversaturation of the detector. Snapshots were taken of the red, green, and bright-field channels and merged using ImageJ (NIH).

### NMR spectroscopy

**Solution NMR samples** hnRNPA2 LC NMR samples were made by diluting protein from 8 M urea into 20 mM MES pH 5.5 with 10% $^2$H$_2$O to a final urea concentration of 150 mM. hnRNPF PLD NMR samples were made by diluting protein from frozen stock (~300 µM) into 20 mM MES pH 5.5 1 mM DTT with 10% $^2$H$_2$O. TOG D1 samples were made by diluting protein from frozen stock into 20 mM MES pH 5.5 1 mM DTT with 10% $^2$H$_2$O. Sample concentrations were estimated using the extinction coefficients calculated by ProtParam.

**Solution NMR experiments** NMR experiments were recorded at 25°C using Bruker Avance III HD NMR spectrometer operating at

850 MHz $^1$H frequency equipped with a Bruker TCI z-axis gradient cryogenic probe. Experimental sweep widths, acquisition times, and the number of transients were optimized for the necessary resolution, experiment time, and signal to noise for each experiment type.

**hnRNPF PLD assignment experiments** Triple resonance assignment experiments were performed on samples of $^{13}$C/$^{15}$N uniformly labeled hnRNPF PLD (conditions: 20 mM MES pH 5.5 1 mM DTT with 10% $^2$H$_2$O). CBCA(CO)NH, HNCACB, HCNO, and HN(CA)CO were recorded with sweep widths 10 ppm (center 4.7) in $^1$H, 20 ppm (center 117) in $^{15}$N, 6.5 ppm (center 173) in $^{13}$C for CO experiments and 56 ppm (center 41) in $^{13}$C for CA/CB experiments using standard Bruker Topspin 3.5 pulse programs with default parameter sets (cbca-conhgp3d, hncacbgp3d, hncacogp3d, hncogp3d). Experiments comprised 84–100, 120, 50, and 3,072 points in the indirect $^{15}$N, indirect $^{13}$Cα/Cβ, indirect $^{13}$CO, and direct $^1$H dimensions, respectively.

**TOG D1 assignment experiments** Triple resonance assignment experiments were performed on samples of $^2$H/$^{13}$C/$^{15}$N uniformly labeled TOG D1 (conditions: 20 mM MES pH 5.5 1 mM DTT with 10% $^2$H$_2$O). HNCA, HN(CO)CA, HNCACB, HN(CO)CACB, HN(CA)CB, HNCO, and HN(CA)CO were recorded with sweep widths 13 ppm (center 4.7) in $^1$H, 25.2 ppm (center 118.65) in $^{15}$N, 12 or 13 ppm (center 173) in $^{13}$C for CO experiments, 60 or 62 ppm (center 42 or 44) in $^{13}$C for CA/CB experiments, and 26 or 28 ppm (center 53) for CA experiments using standard TROSY-based Bruker Topspin 3.5 pulse programs with default parameter sets (trhncagp3d2, trhn-cocacgp3d, trhncacbgp3d, trhncocacbgp3d, trhncacbgp3d, trhncoetgp3d, trhncacogp3d). Experiments comprised 32, 48–64, 96–128, 48, and 1,024 points in the indirect $^{15}$N, indirect $^{13}$Cα, indirect $^{13}$Cα/Cβ, indirect $^{13}$CO, and direct $^1$H dimensions, respectively.

**Paramagnetic relaxation enhancement** Transient intermolecular interactions were probed using paramagnetic relaxation enhancement experiments. The values of the TROSY component of the backbone amide proton transverse relaxation rate constants, $^1$HN $R_2$, were measured as described (Anthis *et al*, 2011) using a two-dimensional TROSY-based experiment at 850 MHz $^1$H frequency for paramagnetic and diamagnetic samples, with 256 and 3,072 total points in the $^{15}$N indirect and $^1$H direct dimensions, corresponding acquisition times of 44 ms and 139 ms, and sweep widths of 34 ppm and 13 ppm centered around 118 ppm and 4.7 ppm, respectively. Each $^1$H$_N$ $R_2$ experiment comprised six interleaved $^1$H$_N$ $R_2$ relaxation delays in this order: 0, 120, 20, 40, 80, and 10 ms.

## C. elegans

### Cloning
The following expression constructs were generated as follows:

- pHA#841: *hrpa-1*p::*hrpa-1*HsLC$^{WT}$::*hrpa-1* 3′UTR (Addgene ID: 139198)
- pHA#842: *hrpa-1*p::*hrpa-1*HsLC$^{D290V}$::*hrpa-1* 3′UTR (Addgene ID: 139199)
- pHA#843: *hrpa-1*p::*hrpa-1*HsLC$^{ΔLC}$::*hrpa-1* 3′UTR (Addgene ID: 139200)
- pHA#847: *mec-4*p::*hrpa-1*mScarlet::*hrpa-1* 3′UTR (Addgene ID: 139201)

- pHA#848: *mec-4*p::*hrpa-1*HsLC$^{WT}$mScarlet::*hrpa-1* 3′UTR (Addgene ID: 139202)
- pHA#849: *mec-4*p::*hrpa-1*HsLC$^{D290V}$mScarlet::*hrpa-1* 3′UTR (Addgene ID: 139203)
- pHA#850: *osm-10*p::FynY531F::*unc-54* 3′UTR (Addgene ID: 139207)
- pHA#851: *osm-10*p::Fyn$^{empty}$::*unc-54* 3′UTR (Addgene ID: 139208)
- pHA#852: *mec-4*p::FynY531F::*unc-54* 3′UTR (Addgene ID: 139209)
- pHA#853: *mec-4*p::Fyn$^{empty}$::*unc-54* 3′UTR (Addgene ID: 139210)

In brief, two fragments containing the *hrpa-1* promoter to the beginning of coding exon 3 (4.4 kb) and the 3′UTR (1.2 kb) were amplified from N2 genomic DNA and assembled using the NEBuilder HiFi DNA Assembly Kit (E2621S) into pBlueScript with a synthesized DNA fragment containing the human hnRNPA2 LC sequence corresponding to the third coding exon of the *C. elegans* gene, codon optimized for *C. elegans* expression. Blunt end site-directed mutagenesis was used to introduce the D290V mutation and a stop codon at the beginning of the 3$^{rd}$ coding exon for ΔLC. To generate *mec-4*p::*hrpa-1* mScarlet plasmids, mScarlet (Bindels *et al*, 2017) was amplified from pmScarlet_C1 (Addgene 85042) and assembled into *hrpa-1*HsLC$^{WT}$ using NEBuilder HiFi DNA Assembly Kit and introducing the D290V mutation by Quickchange. *hrpa-1*mScarlet was generated by cloning the *C. elegans* third exon in place of the optimized human sequence using NEBuilder HiFi DNA Assembly Kit and then cloning in mScarlet as before. The *hrpa-1* promoter was then replaced with the *mec-4* promoter (amplified from *mec-4*p::GFP, from the Driscoll lab) and assembled into *hrpa-1* or *hrpa-1*HsLC$^{WT/D290V}$ using NEBuilder HiFi DNA assembly kit. To generate the Fyn expression constructs, Fyn was amplified from mEos2-FYN2-N-10 (Addgene 57380) and assembled into pBlueScript with either a *mec-4* or *osm-10* promoter and the *unc-54* 3′UTR using the NEBuilder HiFi DNA Assembly Kit. Blunt end site-directed mutagenesis was performed to introduce the Y531F mutation to generate constitutively active Fyn. To generate Fyn$^{empty}$, the Fyn plasmids were cut with HincII and PflM1, gel extracted, and Klenow filled to generate blunt ends and introduce frame shift mutations before ligating.

### Strain construction
See Appendix Table S1 for detailed genotypes of all strains used or generated for this study.

To generate animals expressing mScarlet-tagged HRPA-1 in MEC-4 touch neurons, we generated extrachromosomal arrays by injecting *pha-1(e2123)* animals with the appropriate *mec-4*p::*hrpa-1*mScarlet::*hrpa-1* 3′UTR construct at 5 ng/μl, *mec-4*p::GFP at 10 ng/μl, pBX *pha-1* rescue construct at 100 ng/μl, and 100 ng/μl salmon sperm DNA (to reduce repetitive nature of the array). Animals also expressing a Fyn construct in MEC-4 neurons were generated by injecting *pha-1 (e2123)* with the same pools as above with addition of 5 ng/μl of the appropriate *mec-4*p::Fyn::*unc-54* 3′UTR construct.

To generate *hrpa-1*HsLC animals, we first tried using CRISPR and MosSCI (Zeiser *et al*, 2011) to generate single copy insertions of chimeric *hrpa-1* expressed under its endogenous promoter and 3′UTR. We could not obtain accurate *hrpa-1* homologous recombination events using either method, possibly because of the highly repetitive nature of the endogenous *hrpa-1* gene and the repair construct containing the humanized LC. As such, we generated extrachromosomal arrays by injecting into N2 animals with the *hrpa-1*HsLC constructs at 25 ng/μl, *elt-2*p::GFP at 75 ng/μl, and 100 ng/μl salmon

sperm DNA (to reduce repetitive nature of the array). The resulting extrachromosomal arrays were randomly integrated into the genome by exposing animals to UV irradiation. Integrated transgenes were backcrossed four times, and *hrpa-1(Δ)/balancer* was crossed on to generate the final strains for assay.

To generate animals expressing Fyn constructs in *osm-10* neurons, *pha-1(e2123)* animals were injected with 5 ng/µl *osm-10*p::Fyn::*unc-54* 3′UTR (either Y531F Fyn* or Fyn<sup>empty</sup>), 2.5 ng/µl PCFJ90 (*myo-2*p::mCherry), 100 ng/µl pBX1 *pha-1* rescue construct, and 100 ng/µl salmon sperm DNA. Arrays were selected for at 25°C. Resulting arrays express *myo-2*p::mCherry and were crossed onto integrated *hrpa-1*HsLC arrays with *hrpa-1(Δ)* *(tm781)*, following the array with *myo-2*p::mCherry. Animals were grown at 20°C once crosses with *hrpa-1*HsLC and *hrpa-1(Δ)* were initiated as *pha-1* was not followed.

### Aggregation

Animals were grown at 25°C because all animals are *pha-1(e2123)* with *pha-1* rescue in the extrachromosomal array. Day 1 adult animals (with or without exposure to 22 h 2.5 mM paraquat on plates) were picked into 4 µl M9 and immobilized using NemaGel (Nemametrix). Spots of mScarlet-tagged HRPA-1 (*C. elegans*, HsLC<sup>WT</sup>, or HsLC<sup>D290V</sup>) in the processes of ALM and PLM neurons were counted (63× objective). At least one neuron was counted per animal. Due to the orientation of the animals, often only neurons on one side of the animal could be counted (i.e., 1 ALM and 1 PLM), but sometimes animals were oriented such that 2 neurons of the same type could be counted. There was no difference in number of spots between ALM and PLM. Although they were sometimes visible, PVM and AVM were not scored.

### Fertility

Twelve L4 animals per genotype (not carrying *the tmC25[tmIs1241]* balancer) were singled to NGM plates seeded with 200 µl OP50 (Day 0). Animals were allowed to grow at 20°C for 4 days. On the fourth day, plates were examined for presence of progeny or eggs. Animals were scored as fertile if eggs or progeny were present and not fertile if no eggs could be found on the plate.

### Glutamatergic neurodegeneration

Day 1 adult animals were washed off plates with M9 and incubated with DiD (Fisher DilC18(5) D307) in a microfuge tube as in Perkins *et al* (1986). After 1.5 h, animals were spun down at 10,000 rpm for 30 s and transferred to a regular NGM plate. After 30 min, animals were mounted on 2% (vol/vol) agar pads and immobilized in 30 mg/ml 2-3-butanedione monoxime (BDM, Sigma) in M9 buffer. Fluorescent neuronal cell bodies were visualized and scored for dye uptake under 40× or 63× objectives. There are 4 phasmid neurons per animal, two per side. There are 12 amphid neurons per animal, 6 per side. Neurons were scored as intact if the cell body took up fluorescent dye. Neurons that failed to take up dye could be dead or have degenerated processes. For trials with paraquat stress, animals were pre-exposed to 2.5 mM paraquat for 22 h on plates.

### Cholinergic neuronal death assay

A cholinergic (*unc-17*p::GFP) marker was crossed onto *hrpa-1(Δ)* and *hrpa-1*HsLC<sup>WT</sup>, *hrpa-1*HsLC<sup>D290V</sup>, and *hrpa-1*HsLC<sup>empty</sup> animals.

Homozygous *unc-17*::GFP marker could not be obtained on the *hrpa-1(Δ);hrpa-1*HsLC background, so animals were carefully picked to ensure they were expressing the marker. Day 1 adult animals were mounted on 2% (vol/vol) agar pads and immobilized with 30 mg/ml 2-3-butanedione monoxime (BDM, Sigma) in M9 buffer. Fluorescent neurons were visualized and scored at the microscope for cell death based on loss of neuronal GFP under a 63× objective. Animals missing at least two neurons were considered defective. For trails with paraquat stress, animals were exposed to 2.5 mM paraquat for 22 h on plates.

### RNA extraction and RT–PCR

RNA was extracted from whole animals by solubilizing animals in TRIzol (Life Technologies), adding chloroform, and collecting the aqueous phase containing RNA. RNA was precipitated with isopropanol, washed in 70% ethanol, and dissolved in RNAse- and DNAse-free water. cDNA was made from extracted RNA using the QuantiTect Reverse Transcription Kit (Qiagen). 1 µl of cDNA product was used in a 25 µl PCR with appropriate primers for RT–PCR. Primers are as follows: *hrpa-1*HsLC_F: CTACAACCAGCAGC-CATCTAA, *hrpa-1*HsLC_R: CCTGGACCATAATTTCCTCCTC, *act-1*_F: CAAGGAGTCATGGTCGGTATG, *act-1*_R: TCAATTGGGTACTT-GAGGGTAAG. As Fyn was expressed at low levels in 4-6 neurons, we were unable to confirm expression with RT–PCR.

## Simulations

### Coarse-grained simulations

Coarse-grained simulations on single chains were conducted in cubic boxes with LAMMPS (Plimpton, 1995) software package using our HPS C-α-based model (Dignon *et al*, 2018) coupled to replica exchange molecular dynamics (REMD) (Sugita & Okamoto, 1999) with a temperature list range spanning 150–529K to enhance sampling. HPS treats natural and post-translationally modified residues as single particles. The attractiveness between the particles is scaled based on a common hydrophobicity scale (Kapcha & Rossky, 2014). The hydrophobicity of phosphorylated tyrosine was calculated from ab-initio-derived partial charges (Khoury *et al*, 2013; preprint: Perdikari *et al*, 2020).

### Coarse-grained simulation analysis

Single chain CG simulations of hnRNPA2 LC were conducted for 1µs for a number of phosphorylated tyrosines ranging from 0 (unmodified) to 17 (fully phosphorylated). 10 additional simulations per number of phosphorylation sites ranging from 1 to 16 were carried to calculate the chain dimension of hnRNPA2 LC wild-type and mutants in intermediate phosphorylation states. Only 10 sequences per phosphorylation site were generated because every possible phosphorylation pattern cannot be sampled. The first 100 ns were discarded, and the radius of gyration ($R_g$) was calculated from 5 equal divisions of the equilibrium ensemble at 300 K. $R_g$ uncertainty bars represent SEM from 10 different simulations with the same number of phosphorylated tyrosines randomly placed in different positions. Intramolecular contact maps of hnRNPA2 LC wild-type, D290V, and P298L mutants were constructed based on the average number of intramolecular contacts per frame. The distance cut-off for a contact to be formed is defined as two particles within $2\sigma_{ij}^{1/6}$ Å where $\sigma_{ij}$ is the combined vdW diameter of the two particles (Kim & Hummer, 2008).

### List of simulations and system sizes presented in this work

| System | Force field/model | #Atoms/Particles |
|---|---|---|
| hnRNPA2[190-341]WT | CG HPS model | 152 |
| hnRNPA2[190-341]pY | CG HPS model | 152 |
| hnRNPA2[190-341]D290V (unmodified) | CG HPS model | 152 |
| hnRNPA2[190-341]D290V (pY) | CG HPS model | 152 |
| hnRNPA2[190-341]P298L (unmodified) | CG HPS model | 152 |
| hnRNPA2[190-341]P298L (pY) | CG HPS model | 152 |

## Quantification and statistical analysis

### *C. elegans experiments*

Data collection and analysis were performed by experimenters blinded to genotype. Quantitative results were analyzed using GraphPad Prism 7. ANOVA was used to determine significance and mScarlet assembly assays. Two-tailed t-test was used to determine significance for neurodegeneration assays. A value of $P < 0.05$ was used to establish statistical significance. Error bars in figures represent error or the mean (SEM).

### *Phase separation*

Statistical analysis was performed in Microsoft Office Excel. All data are shown as mean ± standard deviation.

### *NMR Spectroscopy*
*Assignments*

Data were processed with nmrPipe (Delaglio *et al*, 1995) using default linear prediction parameters for either constant time or real-time indirect dimensions and assigned in SPARKY (Lee *et al*, 2015).

### *Chemical shift perturbations and intensity ratio*

Chemical shifts were calculated by subtracting the chemical shifts of a reference peak from the chemical shift of an experimental peak. Intensity ratios were calculated from the height of each peak, and error was propagated from the signal to noise values of each spectrum.

### *Paramagnetic relaxation enhancement values*

Data were processed with nmrPipe (Delaglio *et al*, 1995), apodized with a cosine squared bell function in the $^1$H dimension and a cosine bell function in the $^{15}$N dimension. Best-fit relaxation rates were calculated using least-squares optimization of $^1$H/$^{15}$N peak intensities to a single exponential function. PRE ($\Gamma_2$) rates were obtained from the difference in $^1$H$_N$ $R_2$ rates for the paramagnetic and diamagnetic samples $^1$H$_N$ $R_2^{\text{para}}$ - $^1$H$_N$ $R_2^{\text{dia}}$.

## Data availability

Datasets produced in this study are available in the following databases:

- NMR chemical shift assignments for hnRNPA2 LC WT were previously deposited (BMRB: 27123) (Ryan *et al*, 2018) and can be obtained online from the Biological Magnetic Resonance Database (BMRB http://bmrb.io/).
- NMR chemical shift assignments for hnRNPF PLD (BMRB: 50189) can be obtained from the BMRB (http://www.bmrb.wisc.edu)
- NMR chemical shift assignments for TOG D1 (BMRB: 50258) and can be obtained online from the BMRB (http://www.bmrb.wisc.edu).
- Plasmids generated herein can be found at https://www.addgene.org/Nicolas_Fawzi/.
- Contact In Vivo Biosystems (formerly Nemametrix) to obtain *tdp-1(tgx58)* animals.
- Other *C. elegans* strains generated herein can be obtained from the Caenorhabditis Genetics Center (CGC) (https://cgc.umn.edu/strain/search) or by contacting the Hart lab (anne_hart@brown.edu).

**Expanded View** for this article is available online.

## Acknowledgements

We thank Dr. Vincenzo Venditti for sharing the TROSY-based PRE pulse sequence and Dr. Monica Driscoll for sharing the *mec-4*p::GFP plasmid. Nemametrix made the *tdp-1(tgx58)* strain in collaboration with the Hart Lab (NIA R43AG061978). Research was supported in part by NINDS and NIA R01NS116176 (to NLF and JM), NSF 1845734 (to NLF), NIH R43AG061978 (to ACH), NIGMS R01GM136917 (to JM), and NSF 2004796 (to JM). VHR was supported in part by a Graduate Award from the Robert J. and Nancy D. Carney Institute for Brain Science at Brown University, and Grant F31NS110301 from NINDS, National Institutes of Health. This research is based in part on data obtained at the Brown University Structural Biology Core Facility supported by the Division of Biology and Medicine, Brown University. The use of the high-performance computing capabilities of the Extreme Science and Engineering Discovery Environment (XSEDE), which is supported by the NSF grant TG-MCB-120014, is gratefully acknowledged. Some strains were provided by the Caenorhabditis Genetics Center (CGC), which is funded by NIH Office of Research Infrastructure Programs (P40 OD010440). Some strains were provided by the National Bioresource Project at the Tokyo Women's Medical University School of Medicine funded by the Ministry of Education.

## Author contributions

VHR conceived of the project with input from ACH and NLF. VHR, ACH, and NLF designed all *in vitro* and *C. elegans* experiments. VHR performed and analyzed all *in vitro* and *C. elegans* experiments, with following exceptions: MTN performed TOG D1 assignment experiments and analyzed data/determined the assignments; CFS performed *osm-10*p::GFP *C. elegans* experiments with no stress. TMP, GLD, and JM designed simulation experiments. TMP performed and analyzed all simulation experiments with help from GLD. JL aided with *C. elegans* plasmid cloning, strains, and RT–PCR. VHR, ACH, and NLF wrote the manuscript with input from all authors.

## Conflict of interest

The authors declare that they have no conflict of interest.

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
