## [Review Process File · The EMBO Journal]

Tyrosine phosphorylation regulates hnRNPA2 granule protein partitioning & reduces neurodegeneration

Veronica Ryan, Theodora Perdikari, Mandar Naik, Camillo Saueressig, Jeremy Lins, Gregory Dignon, Jeetain Mittal, Anne Hart, and Nicolas Fawzi

DOI: [10.15252/embj.2020105001](https://doi.org/10.15252/embj.2020105001)

Corresponding author(s): Nicolas Fawzi (nicolas_fawzi@brown.edu) and Anne Hart (anne_hart@brown.edu)

Review Timeline:

Submission Date:	14th Mar 20
Editorial Decision:	28th May 20
Revision Received:	21st Jul 20
Editorial Decision:	20th Aug 20
Revision Received:	14th Oct 20
Accepted:	3rd Nov 20

Editor: Karin Dumstrei

Transaction Report:

Hi Nick,

Thank you for submitting your manuscript to The EMBO Journal and thank you also for your patience with the review of your manuscript. I am very sorry for the delay in getting back to you with a decision, but I have now received the two reports on your study.

As you can see below, the referees find the analysis interesting. However, they also raise a number of different issues with the analysis. Addressing all of them is probably beyond the scope of this manuscript. I am interested in the manuscript and would like to invite you to submit a revised version. I think it would be very helpful to discuss the revisions upfront so that we sort out what needs to be prioritised. We can do so via skype or email whatever works best.

Thank you for the opportunity to consider your work for publication. Looking forward to discussing the revisions further with you

with best wishes

Karin

Karin Dumstrei, PhD
Senior Editor
The EMBO Journal

When assembling figures, please refer to our figure preparation guideline in order to ensure proper formatting and readability in print as well as on screen:
<http://bit.ly/EMBOPressFigurePreparationGuideline>

- a point-by-point response to the referees' comments, with a detailed description of the changes made (as a word file).
- a word file of the manuscript text.
- individual production quality figure files (one file per figure)

- a complete author checklist, which you can download from our author guidelines (<https://www.embopress.org/page/journal/14602075/authorguide>).

- Expanded View files (replacing Supplementary Information)

Further information is available in our Guide For Authors:

The revision must be submitted online within 90 days; please click on the link below to submit the revision online before 26th Aug 2020.

Referee #1:

In this paper, Ryan et al. claim that tyrosine phosphorylation regulates hnRNPA2 partitioning into transport granules (TGs) and reduces neurodegeneration. They claim that other TG components, hnRNPF and ch-TOG, interact weakly with hnRNPA2 but partition specifically into hnRNPA2 liquids. The authors also suggest that tyrosine phosphorylation reduces hnRNPA2 liquid-liquid phase separation (LLPS) and aggregation, and also prevents interactions with hnRNPF and ch-TOG. Finally, a worm model of hnRNPA2 D290V-associated neurodegeneration is established, which exhibits TDP-43 ortholog-dependent glutamatergic neurodegeneration. In this model, expression of a tyrosine kinase that might phosphorylate hnRNPA2 reduces neurodegeneration. This paper is clear, well written, and addresses an area of broad interest. However, there are a number of problems with the paper, which often relate to the physiological relevance of the studies. These problems severely limit the conclusions that can be drawn. They need to be addressed and are summarized below:

1. The authors place too much emphasis on pure protein LLPS/aggregation studies with isolated low-complexity (LC) domains of the proteins of interest. Although this approach is of interest from a physical chemistry perspective, it likely does not reconstitute events that occur in vivo very accurately. Studies with FUS and hnRNPA1 from the Alberti/Hyman and Mittag labs have established that the LC domain only provides a part of the phase-behavior picture for the full-length protein, and that appended domains can have a large impact. Reliance on studies from just the LC domain can be misleading. This issue is further exacerbated by the fact that hnRNPA2 is not truncated in disease. Although many experiments are corroborated with full-length proteins, these findings are relegated to the supplement. These experiments with full-length hnRNPA2 should be moved to the main figures as they are likely of greater physiological relevance. Experiments with just the LC domain of hnRNPA2 should be demoted to the supplement. Moreover, wherever

possible the experiments in Figures 1-4 need to be corroborated with full-length hnRNPA2, FUS, hnRNPF, and ch-TOG. In some cases, it appears that only LC domains have been used, this issue needs to be fixed to make the story more compelling.

2. All of the LLPS/aggregation studies are highly qualitative and rely on a few fluorescence/DIC images. Often we only see one droplet in an image. To provide a more accurate picture of events, these assays must be made quantitative. What is the area covered by droplets/aggregates, what is the droplet/aggregate number per unit area, what is the average size of droplets/aggregates? Without this type of information, the data are difficult to assess. Likewise, alternative readouts of LLPS and aggregation would also be valuable to corroborate findings/conclusions. For example, light scattering and quantitating the amount of phase separated/aggregated protein would be very important to validate conclusions drawn. This is done for some constructs in Fig. S1J & S3F, but much more of this is needed.

3. What is the K_d between hnRNPA2 LC and TOG D1? It is stated that this is a weak interaction, but no measure of interaction strength is provided.

4. It is not clear what proportion of tyrosines are phosphorylated in pY hnRNPA2 FL or which residues. Thus, it is difficult to assess these experiments. It is also unclear which tyrosines are getting phosphorylated in the pY hnRNPA2 LC domain. The authors should provide evidence that hnRNPA2 does actually get phosphorylated on tyrosine residues in cells. They should also define under which circumstances does this occur? This same concern extends to the serine phosphomimetic constructs. Is hnRNPA2 actually ever phosphorylated at these sites in cells and, if so, under what circumstances? The experiments ought to be guided by the hnRNPA2 phosphoforms that actually occur in cells.

5. It is not surprising that replacing S/T/Y with E reduces prion-like character as assessed by PLAAC. E is disfavored in prion domains as originally defined by Alberti and colleagues.

6. The rationale for expressing a chimeric human/worm protein in *C. elegans* is unclear as now the interdomain interactions have been completely altered, which could affect LLPS and aggregation behavior (as noted in point 1). The use of this chimera weakens the study. It would be more compelling to generate a worm model expressing human hnRNPA2 or disease-linked variants. In this way, one can be certain that any findings are directly relevant to the human protein and disease-linked variants. *C. elegans* models of this kind have been valuable in studying several other neurodegenerative disorders.

7. Much is made of how hnRNPA2 is a component of transport granules. However, it is also a component of stress granules. In the worm model, HRP1 granules are only induced by stress, which would seem to make it likely that these are stress granules and not transport granules (are transport granules ever found in *C. elegans*?). This disconnect between the two parts of the story is jarring and more work needs to be done to define what the granules are in worm (it seems most likely that these are stress granules, which could be corroborated by staining for several markers). Increased recruitment of hnRNPA2-D290V to stress granules is already known, which makes these findings unsurprising.

8. Expression of Fyn kinase is found to decrease formation of granules by HRP-1HsLCD290V but not HRP-1HsLCWT in worm. However, it is unclear why WT and mutant forms would respond differently. Several controls are missing from these studies. It should be demonstrated that HRP-1HsLCD290V is indeed phosphorylated by Fyn in worm. A kinase-dead control is missing.

Expression of a mutant form HRP-1HsLCD290V that lacks the tyrosines that would be phosphorylated by Fyn (e.g. Y to F or Y to W variants) should also be included. Without these controls it is not clear whether the effects are really due to direct tyrosine phosphorylation of HRP-1HsLCD290V by Fyn or whether the effects are due to some other indirect of Fyn, e.g. phosphorylation of other granule scaffolds or some other effect such as phosphorylation and inactivation of a chaperone for HRP-1HsLCD290V or inactivating phosphorylation of TDP-43. These issues also apply to experiments where Fyn is found to rescue neurodegeneration due to HRP-1HsLCD290V (Fig. 6).

9. The connection with TDP-43 is intriguing and loss of TDP-43 function seems to rescue HRP-1HsLCD290V-mediated neurodegeneration. However, this finding seems counter to situations of human disease where loss of TDP-43 function in the nucleus is likely a key driver of neurodegeneration. How is TDP-43 function driving neurodegeneration?

10. Fyn kinase seems to rescue HRP-1HsLCD290V-mediated neurodegeneration, but it is not clear whether this is due to phosphorylation of TDP-43 (leading to loss of TDP-43 function), HRP-1HsLCD290V, or both. This issue needs to be clarified.

11. Is HRP-1HsLCD290V-mediated neurodegeneration accompanied by HRP-1HsLCD290V aggregation or TDP-43 aggregation? The answer to this question is important so that we can understand whether the worm model phenocopies these aspects of human disease.

12. In the discussion, the authors note 'As such, it is possible that aberrant assembly or aggregation of hnRNPA2 driven by disease mutation is toxic and the presence of these assemblages directly correlates with neurodegeneration.' Data from the Taylor lab has already established that this is the case in fly.

13. Fig. 7 does not convey a very clear model and should be revised more clearly explain the advances made in this paper.

Referee #2:

In the manuscript titled "Tyrosine phosphorylation regulates hnRNPA2 granule protein partitioning & reduces neurodegeneration", Dr. Fawzi and colleagues describe the how the liquid-liquid phase separation of low hnRNPA2, a transport granule component, is influenced by tyrosine phosphorylation, for example by Fyn kinase. Furthermore, the effect on partitioning of other two granule proteins, hnRNPF and ch-TOG, is studied as well. Using an elaborated set of genetic constructs of the low complexity domain of hnRNPA2 and of FUS, they show - in vitro by microscopy and NMR - that weak interactions based on charge and certain residues, like arginine and p-tyrosine, play a regulatory role for hnRNPA2 LLPS and granule component co-partitioning. Furthermore, tyrosine phosphorylation also inhibits/delays LLPS and aggregation of disease associated hnRNPA2 mutants in vitro, and reduces the number of hnRNPA2 accumulations in glutamatergic neurons in C elegans; C elegans lines expressing the nematode hnRNPA2 orthologue *hrp-1* with with chimera LCDs were used to visualize the accumulation of hnRNPA2 under stress, to study the neurodegenerative effect of these accumulations, to show the beneficial effect of

tyrosine phosphorylation by Fyn in vivo, and to show that TDP43 orthologue tdp-1 might be involved in neurotoxicity.

In general, reports very interesting findings. The authors use a broad spectrum of techniques to emphasize the relevance of PTMs for granule biology, and also addresses this question in vivo in *C. elegans*; the translation into a living model system can be seen as an important contribution and effort, which we have to appreciate in the LLPS field. All experiments are well designed and contribute to the major questions, however, some of the *C. elegans* experiments appear to be underpowered (details see comments below), and some in vitro experiments, that would support the author's conclusions, are missing. The text is well written but some aspects are hard to follow due to complicated nomenclature, wrong figure panel order, incomplete statement of buffer conditions in text or figure legends, and missing statistical details.

In the following you find my major, minor, and additional comments.

Major points:

General comments:

For transparency and understanding, domain structures with residue numbering for all proteins and artificial constructs should be shown in the appropriate figures; for hnRNPA2, hnRNPF, and TOG, and for FUS constructs with changed residues. It would also be helpful to show net charge/surface charge, and if possible charge pattern for all constructs.

-

Salt and protein concentrations and pH should be given consequently as numbers in the text and the figures, and not as "physiological"; this can mean a wide range. This would make it much easier for the reader who otherwise has to always go back to the methods part and dig out the right experimental setup to see the actual salt, pH, protein, etc conditions.

Specific comments:

Which salt and protein concentrations were used in the NMR titration experiments of hnRNPA2 LC and hnRNPF PDL and TOG D1? Was this under co-LLPS conditions or in conditions of monomeric interactions? How do the spectra of these two interaction scenarios (co-condensation vs monomeric interactions) compare?

-

The authors state (in the context of the NMR data in Figure S1E) that hnRNPA2 LC does not phase separate at low salt concentrations, and that hnRNPF PDL is capable of inducing condensation under these conditions. This should be shown by microscopy and the exact conditions (salt, pH, protein conc.) should be given.

-

Supplemental Figures S1, S2, and S3 are too packed and the displayed images are therefore too small to judge the presence of condensates in some of them. I suggest to split these figures each into 3 separate figures, and remove most of the TEV- data, since it is obvious from the basic constructs like hnRNPA2 LC +/- TEV that addition of TEV initiates LLPS.

-

How do you put the changes in hnRNPA2 LC LLPS into context of the previously shown PRMT1 induced methylation of hnRNPA2 and its effects on LLPS? Does any PTM cancel hnRNPA2 LLPS, or are there some that enhance LLPS? please, at least, discuss this.

--

If pY is "selectively" canceling LLPS and co-LLPS of other proteins, how about other

phosphorylations? Is the presence of phosphate groups per se influencing the granule assembly? You showed pseudo-phosphorylation of serines, but what about using another serine/threonine kinase that actually phosphorylates these residues?

What happens if you phosphorylate the clients hnRNPF PDL or TOG D1, do you see a similar or opposite effect?

Can you reconstitute LLPS and co-LLPS by dephosphorylation of pY hnRNPA2 LC?

--

The authors speculate that "hnRNPA2 LC self-interactions between phospho-tyrosine and arginine outcompete the weak interactions with hnRNPF PDL, thus preventing its partitioning." This is an interesting hypothesis that the authors could attempt to test, e.g. by experiments, in which a) hnRNPA2 LC + hnRNPF PDL are mixed to co-condensate, and then pYhnRNPA2 LC is added to see if hnRNPF PDL gets depleted from the condensates. They could also try to outcompete against hnRNPF PDL with soluble pY.

--

True granules contain more than 2 components; what happens if you mix hnRNPA2, hnRNPF, and TOG D1? Do they compete in partitioning? And how is that regulated by RNA, a major and essential constituent of the granules?

--

It was observed that glutamines in the positions of tyrosine in hnRNPA2 LC increase the "prion-like" character. Please discuss why that might happen.

--

In addition: across the manuscript, the term "prion-like" is used without definition which characteristics are hiding behind this term. The authors have to at least once in the beginning give a definition what they mean with "prion-like" (LLPS to aggregate transition????) and where this originates from. Real prion-behavior includes structural conversion as well as cell transmission, I guess that's not what you are talking about. I suggest to define or simply describe what is changing instead of naming it "prion-like".

It is not defined and explained what PRE and PLAAC are, and what they measure.

--

In vivo studies:

Please proof - biochemically for protein and/or RNA levels - the successful knock-down and expression of hrp-1 and Fyn in the C elegans lines.

-

What is the phosphorylation status of HRP-1 with and without expression of Fyn?

--

Do the worms show neuronal loss after stress? In Figure S6F, no difference in GFP+ neuronal numbers is shown in absence of stress. What is the number of GFP+ neurons in C elegans after stress application?

--

Do the lines that carry mScarlet-containing constructs show neurodegeneration or neuronal loss?

--

The experiments showing dye uptake in neurons of WT and D290V HsLCmScarlet lines - presented in Figure 5F+G - seem underpowered with n=3 repetitions. They should be repeated at least 2-3 more times. Additionally, a statistical comparison between stressed and no-stress conditions for all groups is missing.

--

The findings about TDP-43-related neurodegeneration in C elegans are interesting but need a lot more experimental elaboration and - in the current form - seem out of the scope of this manuscript - at least in their current state and as presented: they open a lot of obvious questions about the

relationship between hnRNPA2 and TDP43, which are not addressed.

The findings are also not discussed at all, and seem to not contribute much to the "story".

I suggest to take the TDP43 data out from this manuscript and focus on phosphorylation (and other PTMs) and the effect on granule assembly. If left in the ms, the authors should address at least the following questions regarding hnRNPA2:TDP-43 interaction in neurodegeneration: 1) the authors previously showed that hnRNPA2 and TDP43 can co-phase separate; how is co-LLPS effected by tyrosine phosphorylation? 2) Do TDP-43 and hnRNPA2 co-localize in stress-induced neuronal "spots", in C elegans and in mammalian neurons or at least mammalian neuronal cells? 3) How does Fyn expression alter TDP43 PTMs, phase separation, aggregation, and nuclear:cytosolic localization? It needs to be shown biochemically that TDP-1 is knocked-out in the worms, as well as if the localization of TDP-1 is changed upon stress and Fyn expression, in both the WT and mutant hnRNPA2 LC expressing lines.

Minor points:

- hnRNPA2 is multiple times written as hnRNAP2
- Page 2 lane 9: ...these these... > ...these...
- Page 4 lane 1: ... two these ... > ...these two...
- Page 4 lane 20: ...hnRNPA2 and hnRNPF... > ... hnRNPA2 LC and hnRNPF PLD...
- Page 5 lans 1: ... but not known to be ... > Do you mean "not known to be" or "known to not be"
- Page 5 lane 20: ..."hnRNPA2 LC-like" patter of charges residues... > do you mean "charge pattern" or "net charge"?
- The authors state specificity of hnPNPF PDL co-condensation with hnRNPA2 LC. However, they also show that exchange of a few charged residues in FUS can enable hnRNPF co-LLPS. - I would rather interpret this as non-specific co-LLPS of hnRNPF that is mostly dependent on protein charge but not sequence specificity, since the authors showed that hnRNPF PDL can co-condensate with [FUS LC + Rs] because of charge. Later in the manuscript, when tyrosine phosphorylation is analyzed, it is shown that charge change by phosphorylation is "not responsible for specificity of interactions with hnRNPA2 LC" (Page 8). "...not exclusively responsible..." Would be more appropriate here I think. I suggest to be more careful with the chosen phrasing and the interpretation. However, it also seems like a logical conclusion that protein conformations, and thus interactions, are different in the presence of phosphorylated tyrosines (or other residues) compared to glutamates due to the bulky phosphate groups that present three negative charges in close proximity and thereby enable very precise, oriented electrostatic interactions.
- In most cases the authors talk about "tyrosine" and "arginine"; this should be changed to "tyrosine residues" or "tyrosine", adequately for arginine and other residues like serine etc.
- Page 5 lane 22: ..."FUS LC-like" depletion of charged residues...> I suggest to remove the expression "FUS LC-like" or replace like depletion of charged residues, similar to FUS LC,
- Page 6 lane 10: ...we hypothesized that the 1st domain of ch-TOG would also partition into hnRNPA2 droplets. It is unclear what is the rationale behind this hypothesis, in other words where this idea comes from and why not other domains are tested. It would help if the authors would show TOG domain structure and state why D1 likely partitions.
- Page 6 lane 25:of between.... > ...of...
- Page 9 lane 18: ...reduce prion-like... > ...reduce the prion-like....
- Page 10 lane 15: the authors should give the total number of residues for hnRNPA2 LC and its orthologues and at least mention, better comment on, the reduced number of serines and enhanced number of glutamines in Drosophila and C elegans.
- Page 10 lane 17: ...HRP-1 function....> Do the authors mean HRP-1 condensation capability? They are not testing the actual protein function here

- Page 10 lane 21: please specify "low salt conditions" and "physiological salt (concentration)". - Furthermore, the pH is different between the conditions in reported; does the low pH5.5 lead to aggregation of HRP-1?
- Page 10 lane 22/23: ...self-assembly...> what is self-assembly of a LCD? please define more precisely, or use self-association, which is a more broad expression compared to usually well organized self-assembly.
- Page 10 lane 23: ...we created C.e. strains expressing a chimeric HRP-1 protein containing most of the human hnRNPA2 LC domain. > it is not clear what is the rational behind these lines; the authors need to explain, what they expect to gain from these lines, and why they chose the chimera as they did.
- Figure S5A: needs to explain better, which residues of hnRNPA2 LC were transferred and to which location in the hrp-1 LC sequence
- Page 11 lane 1: I guess the sentence should be: ...if HRP-1 assembles into stress granules, transport granules, or aggregates, all of them together are referred to herein as "spots".
- Page 12 lane 2: ...No overt defects were observed... > please specify: morphologically, behaviorally?
- Page 12 lane 4: ...disease models... > ... disease models of C elegans...
- Page 13 lane 13: please give name of GFP line please.
- Page 13 lane 16: it is absolutely unclear what "+" means the text. This is also true in the corresponding figure panels. The authors need to clearly state which line or treatment is labeled with "+".
- Page 13, paragraph starting lane 24: It seems to me that this paragraph would have a better place further up, right after the authors report the data from the mScarlet constructs in the context of Fyn?
- Page 14 lane 11 (discussion part): ...distinct prion-like domains. > I suggest to change to distinct low-complexity domains.
- Page 15 lane 16:...assemblages... > ...assemblies...
- Page 15 lane 19: can you comment on the role of Fyn in other neurodegenerative diseases? Would it also make sense to boost Fyn in other disease contexts?

Figure 1:

- Add protein domain and construct schematic for hnRNPA2, hnRNPF, and FUS
- The data for the FUS control in (B) could go in supplemental and instead show the quantification as in Figure S1J)
- Add imaging data for FUS LC(40uM)+ hnRNPF PDL (40uM) (no co-LLPS?) to panel D. Also, for easy readability, please give protein concentrations in all figure panels.
- Panel c): unclear what the different colored bars are; please describe in the legend. What is "physiological pH"? What are the negatively charged residues?

Figure 3:

- Panel c): nomenclature different than in ms text: pYhnRNPA2 instead of phnRNPA2

Figure 4:

- Panel b): what is PLAAC and what does it measure?

Figure 5:

- Panel d): which worm genotype is shown?
- Panel e): marker size in graph are not uniform
- Panel f+g): how many animals and neurons did you analyze? From the data point distribution and SD it looks like these experiments are underpowered and more animals and/or repetitions of the experiment have to be added. Also problematic: the way that the conditions are compared. why is the HsLcWT without Fyn not giving a difference between stress and no-stress as before reported in panel E)? Were those conditions statistically compared at all in this experiment?

Figure 7:

- The figure legend is poor. What are the red crosses standing for? In the in vivo part, what is this supposed to be?

Figure S1:

- Figure is too crowded, too small images. Please split Figure in 2-3 figures: e.g. A+B; C+D+E+F; G+H+I+J

- Same is true for Figure S2 and S3

Figure S6:

- Panel F: what does "ASH" mean? What about the number of GFP+ neurons in stressed animals?

Additional comments:

The authors use an overly complicated nomenclature of the used constructs; I suggest to find another shorter but intuitive way to name the constructs, e.g. "hnRNPA2*" instead of "hnRNPA2 LC no charge" and "hnRNPA2*R" instead of "hnRNPA2 LC no charge with R", or similar; R to K could be written as R/K. This will make the text much better readable and understandable.

--

At many places in the manuscript, the order of Figure panel is not in order; this has to be corrected by either text modification or Figure rearrangement.

--

Often poor figure legends that lack description of experimental setup, conditions, statistics.

We thank the reviewers for their careful examination of our data and paper. Below is our point-by-point response to the points that were raised.

Referee #1:

In this paper, Ryan et al. claim that tyrosine phosphorylation regulates hnRNPA2 partitioning into transport granules (TGs) and reduces neurodegeneration. They claim that other TG components, hnRNPF and ch-TOG, interact weakly with hnRNPA2 but partition specifically into hnRNPA2 liquids. The authors also suggest that tyrosine phosphorylation reduces hnRNPA2 liquid-liquid phase separation (LLPS) and aggregation, and also prevents interactions with hnRNPF and ch-TOG. Finally, a worm model of hnRNPA2 D290V-associated neurodegeneration is established, which exhibits TDP-43 ortholog-dependent glutamatergic neurodegeneration. In this model, expression of a tyrosine kinase that might phosphorylate hnRNPA2 reduces neurodegeneration. This paper is clear, well written, and addresses an area of broad interest. However, there are a number of problems with the paper, which often relate to the physiological relevance of the studies. These problems severely limit the conclusions that can be drawn. They need to be addressed and are summarized below:

1. The authors place too much emphasis on pure protein LLPS/aggregation studies with isolated low-complexity (LC) domains of the proteins of interest. Although this approach is of interest from a physical chemistry perspective, it likely does not reconstitute events that occur in vivo very accurately. Studies with FUS and hnRNPA1 from the Alberti/Hyman and Mittag labs have established that the LC domain only provides a part of the phase-behavior picture for the full-length protein, and that appended domains can have a large impact. Reliance on studies from just the LC domain can be misleading. This issue is further exacerbated by the fact that hnRNPA2 is not truncated in disease. Although many experiments are corroborated with full-length proteins, these findings are relegated to the supplement. These experiments with full-length hnRNPA2 should be moved to the main figures as they are likely of greater physiological relevance. Experiments with just the LC domain of hnRNPA2 should be demoted to the supplement. Moreover, wherever possible the experiments in Figures 1-4 need to be corroborated with full-length hnRNPA2, FUS, hnRNPF, and ch-TOG. In some cases, it appears that only LC domains have been used, this issue needs to be fixed to make the story more compelling.

We appreciate the reviewer's concern that we have placed primary emphasis in this study on the low complexity domain. After discussion with the editor, we have left the FL mixing data in the supplement but have updated the text to reflect that the domains are being used to model the kinds of interactions the proteins are making. This decision was made because the point of the manuscript is not to establish the low complexity domain droplets themselves as faithful models of the granules – the point of the manuscript is to examine in detail the interactions mediated by the disordered domains. We considered corroborating all the experiments using full length protein, but we did not anticipate that the specificity of hnRNPF and TOG1 for hnRNPA2 LC vs FUS LC would translate to the full-length proteins because full-length FUS has arginine residues. Therefore, to test this hypothesis, we performed mixing experiments with FUS FL and found that both TOG D1 and hnRNPF were able to partition into FUS FL droplets (likely due to the presence of the highly charged RGG domains). We have added comments in the text emphasizing again the point of the study is to understand that molecular interactions mediated by the low complexity domains and how the low complexity domains of hnRNPA2 and FUS are not identical – a fact that would be lost using full-length proteins. Finally, it is important to note that even using the full-length protein alone does not reconstitute the many components and complex environment of RNA granules in cells. We underscore again the importance of understanding the molecular interaction specificity of the LC domains because so many claims

are made regarding their behavior in the literature.

2. All of the LLPS/aggregation studies are highly qualitative and rely on a few fluorescence/DIC images. Often, we only see one droplet in an image. To provide a more accurate picture of events, these assays must be made quantitative. What is the area covered by droplets/aggregates, what is the droplet/aggregate number per unit area, what is the average size of droplets/aggregates? Without this type of information, the data are difficult to assess. Likewise, alternative readouts of LLPS and aggregation would also be valuable to corroborate findings/conclusions. For example, light scattering and quantitating the amount of phase separated/aggregated protein would be very important to validate conclusions drawn. This is done for some constructs in Fig. S1J & S3F, but much more of this is needed.

We thank the reviewer for this comment. Because the issues arising from quantifying concentration from fluorescent dyes (see Transparent Peer Review for Wegmann et al EMBO J 2018 for example), we did not quantify these images. In addition, droplet size, area, and number are all strongly dependent on time between initiation of LLPS and imaging, and we are skeptical in the value of quantification of parameters that are so sensitive and are not “thermodynamic” quantities. Rather, here we focus on looking for qualitative differences between partitioning (did the client go into the scaffold droplet or not? yes or no) and have edited the text to more accurately reflect this.

3. What is the Kd between hnRNPA2 LC and TOG D1? It is stated that this is a weak interaction, but no measure of interaction strength is provided.

We thank the reviewer for pointing out this oversight, the estimated Kd is much greater than 100 μ M based on the NMR spectroscopy experiments and we have added this to the text.

4. It is not clear what proportion of tyrosines are phosphorylated in pY hnRNPA2 FL or which residues. Thus, it is difficult to assess these experiments. It is also unclear which tyrosines are getting phosphorylated in the pY hnRNPA2 LC domain. The authors should provide evidence that hnRNPA2 does actually get phosphorylated on tyrosine residues in cells. They should also define under which circumstances does this occur? This same concern extends to the serine phosphomimetic constructs. Is hnRNPA2 actually ever phosphorylated at these sites in cells and, if so, under what circumstances? The experiments ought to be guided by the hnRNPA2 phosphoforms that actually occur in cells.

*We thank the reviewer for this comment. We note that hnRNPA2 is known to be tyrosine phosphorylated by Fyn at sites of local translation (White 2008) and has found to be tyrosine phosphorylated in several proteomic studies (PhosphoSite) at 13 different tyrosines (of 17 total tyrosines in the LC). It is likely that other positions are modified, but identification of the remaining sites may be difficult due to the repeating sequence and incomplete coverage of phosphoproteomics in low complexity domains (see our work Monahan et al. EMBO J 2017 where we examined this in detail for serine/threonine phosphorylation of FUS). Determining the specific phosphoproteomforms is extremely challenging and given the low complexity sequence, phosphorylation pattern is more likely to be stochastic than to have specificity (most likely, the specificity is encoded at the protein level, in binding to Fyn or not, but not in the position level, of where in the LC sequence the phosphorylation occurs). Hence, it is well established in the literature that hnRNPA2 is phosphorylated on tyrosine positions and that Fyn is responsible for at least some of this phosphorylation. As we describe below, given that we are expressing Fyn in only a few neurons, it is not feasible to collect enough hnRNPA2 chimeric protein for western blot or phosphoproteomic mass spectrometry in *C. elegans*. We think our recombinant phosphorylated hnRNPA2 LC contains a mixture of phosphorylated tyrosines (e.g., 1, 2, 3, 4 in molecule one, 2, 5, 7, 9, 10, 13 in molecule 2, etc). Hence, we suggest that our phosphorylated*

hnRNPA2 LC is a good model for hnRNPA2 phosphoforms that occur in cells – and it is certainly the best model to date.

5. It is not surprising that replacing S/T/Y with E reduces prion-like character as assessed by PLAAC. E is disfavored in prion domains as originally defined by Alberti and colleagues. *We agree with the reviewers and this is precisely the point we were trying to make. We have clarified this in the text.*

6. The rationale for expressing a chimeric human/worm protein in *C. elegans* is unclear as now the interdomain interactions have been completely altered, which could affect LLPS and aggregation behavior (as noted in point 1). The use of this chimera weakens the study. It would be more compelling to generate a worm model expressing human hnRNPA2 or disease-linked variants. In this way, one can be certain that any findings are directly relevant to the human protein and disease-linked variants. *C. elegans* models of this kind have been valuable in studying several other neurodegenerative disorders.

*We appreciate the concerns of the reviewer, but we disagree with this perspective. There is minimal evidence for interdomain interactions of hnRNPA2 (or the related A1) and no evidence for tight interaction. The RRM s are well conserved between *C. elegans* and humans and this substitution of LC is a minimal change – by changing the RRM s we would have to evaluate if the difference observed is due to changes in the RRM s. Here we are confident that the difference arises only from the LC in the chimeric model.*

7. Much is made of how hnRNPA2 is a component of transport granules. However, it is also a component of stress granules. In the worm model, HRP1 granules are only induced by stress, which would seem to make it likely that these are stress granules and not transport granules (are transport granules ever found in *C. elegans*?). This disconnect between the two parts of the story is jarring and more work needs to be done to define what the granules are in worm (it seems most likely that these are stress granules, which could be corroborated by staining for several markers). Increased recruitment of hnRNPA2-D290V to stress granules is already known, which makes these findings unsurprising.

*We thank the reviewer for this comment. We agree with the reviewer that the manuscript was written to emphasize hnRNPA2 transport granules despite the fact that we do not extensively characterize granules of hnRNPA2 in cell. We do note that *C. elegans* likely have transport granules as mRNAs are found in synapses (Arey 2019 Scientific Reports), RNA binding proteins are found in neuronal dendrites (Antonacci 2015 G3), and TIA-1 positive liquid-like granules form in regenerating axons in *C. elegans* (Andrusiak 2019 Neuron). Discriminating between stress granules, transport granules, aggregates, and other entities would require studies beyond the scale of this analysis. We take a conservative approach of calling the accumulations that we do observe (in the neuronal processes) “spots” to avoid misleading the reader. Staining for worm proteins is tricky (especially proteins with IDRs) as there is often not sufficient conservation of the epitopes (from vertebrates) against which well-validated antibodies were raised and commercially available reagents often show spurious cross-reactivity in *C. elegans* due to filarial infections in animals used to raise sera (i.e. one usually can't use antibodies against mammalian proteins in worms).*

8. Expression of Fyn kinase is found to decrease formation of granules by HRP-1HsLCD290V but not HRP-1HsLCWT in worm. However, it is unclear why WT and mutant forms would respond differently. Several controls are missing from these studies. It should be demonstrated that HRP-1HsLCD290V is indeed phosphorylated by Fyn in worm. A kinase-dead control is missing. Expression of a mutant form HRP-1HsLCD290V that lacks the tyrosines that would be phosphorylated by Fyn (e.g. Y to F or Y to W variants) should also be included. Without these

controls it is not clear whether the effects are really due to direct tyrosine phosphorylation of HRP-1HsLCD290V by Fyn or whether the effects are due to some other indirect of Fyn, e.g. phosphorylation of other granule scaffolds or some other effect such as phosphorylation and inactivation of a chaperone for HRP-1HsLCD290V or inactivating phosphorylation of TDP-43. These issues also apply to experiments where Fyn is found to rescue neurodegeneration due to HRP-1HsLCD290V (Fig. 6).

We thank the reviewer for this comment. Unfortunately, it is incredibly difficult to obtain sufficient protein from C. elegans for pull downs or LC MS-MS studies, particularly when the protein of interest is only expressed in 2-4 neurons. Further, changes to the LC sequence of hnRNPA2 alter phase separation and aggregation propensity (see Figure 1F, 4A, S7B) and tyrosine to phenylalanine mutations drastically alter FUS LC LLPS and induced aggregation (see Murthy et al 2019 NSMB) and hence all tyrosines cannot be mutated to phenylalanine (indeed, tyrosine is highly conserved here). Similarly, tryptophans appear to provide a much stronger contribution to LLPS of the related protein TDP-43 compared to tyrosine residues (Li et al. Journal of Biological Chemistry, 2018 doi: 10.1074/jbc.AC117.001037), hence mutating all 17 tyrosine to tryptophan is not conservative enough to expect nothing will be perturbed except phosphorylation. We therefore did not attempt to perform further experiments along this line.

We point out that Fyn is not known to have a scaffolding interaction that a “kinase-dead” version of Fyn would address beyond the Fyn-empty herein (where no Fyn protein is expressed). On the other hand, we agree with the reviewer that the effect of active Fyn may be indirect (e.g. due to phosphorylation of other proteins) and we have included this caveat in the revised manuscript.

9. The connection with TDP-43 is intriguing and loss of TDP-43 function seems to rescue HRP-1HsLCD290V-mediated neurodegeneration. However, this finding seems counter to situations of human disease where loss of TDP-43 function in the nucleus is likely a key driver of neurodegeneration. How is TDP-43 function driving neurodegeneration?

We agree with the reviewer that this was not well explained. We have revised this text to better describe how the tdp-1 data enables us to further characterize the model and clearly demonstrate effects of the mutant LC that cannot be accounted for by loss of function.

Therefore, tdp-1 data are not intended for comparison with human disease. However, a similar network of hnRNP proteins is present across animals and their interactions have begun to be mapped out. Nor do we intend here to characterize how tdp-1 drives neurodegeneration – we agree it is interesting but it is not in the scope of this work. In our opinion, these results here are clear and will be of interest to the community.

10. Fyn kinase seems to rescue HRP-1HsLCD290V-mediated neurodegeneration, but it is not clear whether this is due to phosphorylation of TDP-43 (leading to loss of TDP-43 function), HRP-1HsLCD290V, or both. This issue needs to be clarified.

We thank the reviewer for this comment and think it was addressed in points 8 and 9. We note that tdp-1 and TDP-43 do not have significant number of tyrosine residues (only 1 in TDP-43 CTD and 6 in tdp-1 compared to 17 in hnRNPA2).

11. Is HRP-1HsLCD290V-mediated neurodegeneration accompanied by HRP-1HsLCD290V aggregation or TDP-43 aggregation? The answer to this question is important so that we can understand whether the worm model phenocopies these aspects of human disease.

We agree with the reviewer that this is an excellent question but unfortunately we don't currently have a reliable way to visualize TDP-43 aggregation. We feel that experiments probing this important question are out of the scope of this paper.

12. In the discussion, the authors note 'As such, it is possible that aberrant assembly or

aggregation of hnRNPA2 driven by disease mutation is toxic and the presence of these assemblages directly correlates with neurodegeneration.' Data from the Taylor lab has already established that this is the case in fly.

We thank the reviewer for this comment and cite the work by J Paul Taylor and coworkers (Kim et al Nature 2013) in this place in the revised manuscript.

13. Fig. 7 does not convey a very clear model and should be revised more clearly explain the advances made in this paper.

We thank the reviewer for this comment and have modified the figure.

Referee #2:

In the manuscript titled "Tyrosine phosphorylation regulates hnRNPA2 granule protein partitioning & reduces neurodegeneration", Dr. Fawzi and colleagues describe the how the liquid-liquid phase separation of low hnRNPA2, a transport granule component, is influenced by tyrosine phosphorylation, for example by Fyn kinase. Furthermore, the effect on partitioning of other two granule proteins, hnRNPF and ch-TOG, is studied as well. Using an elaborated set of genetic constructs of the low complexity domain of hnRNPA2 and of FUS, they show - in vitro by microscopy and NMR - that weak interactions based on charge and certain residues, like arginine and p-tyrosine, play a regulatory role for hnRNPA2 LLPS and granule component co-partitioning. Furthermore, tyrosine phosphorylation also inhibits/delays LLPS and aggregation of disease associated hnRNPA2 mutants in vitro, and reduces the number of hnRNPA2 accumulations in glutamatergic neurons in C elegans; C elegans lines expressing the nematode hnRNPA2 orthologue hrp-1 with chimera LCDs were used to visualize the accumulation of hnRNPA2 under stress, to study the neurodegenerative effect of these accumulations, to show the beneficial effect of tyrosine phosphorylation by Fyn in vivo, and to show that TDP43 orthologue tdp-1 might be involved in neurotoxicity.

In general, reports very interesting findings. The authors use a brought spectrum of techniques to emphasize the relevance of PTMs for granule biology, and also addresses this question in vivo in C elegans; the translation into a living model system can be seen as an important contribution and effort, which we have to appreciate in the LLPS field. All experiments are well designed and contribute to the major questions, however, some of the C elegans experiments appear to be underpowered (details see comments below), and some in vitro experiments, that would support the author's conclusions, are missing. The text is well written but some aspects are hard to follow due to complicated nomenclature, wrong figure panel order, incomplete statement of buffer conditions in text or figure legends, and missing statistical details.

In the following you find my major, minor, and additional comments.

Major points:

General comments:

For transparency and understanding, domain structures with residue numbering for all proteins and artificial constructs should be shown in the appropriate figures; for hnRNPA2, hnRNPF, and TOG, and for FUS constructs with changed residues. It would also be helpful to show net charge/surface charge, and if possible charge pattern for all constructs.

We thank the reviewer for this comment and have added figures as requested.

-

Salt and protein concentrations and pH should be given consequently as numbers in the text and the figures, and not as "physiological"; this can mean a wide range. This would make it much easier for the reader who otherwise has to always go back to the methods part and dig out the right experimental setup to see the actual salt, pH, protein, etc conditions.

We thank the reviewer for this suggestion and have checked that this information was clear in the figure legends and text. We agree that "physiological" is insufficiently descriptive.

Specific comments:

Which salt and protein concentrations were used in the NMR titration experiments of hnRNPA2 LC and hnRNPF PLD and TOG D1? Was this under co-LLPS conditions or in conditions of monomeric interactions? How do the spectra of these two interaction scenarios (co-condensation vs monomeric interactions) compare?

We thank the reviewer for this comment. The NMR was done under monomeric conditions (absence of salt, specifically 20 mM MES pH 5.5 – pH 5.5 is chosen only to improve NMR spectra – see Ryan et al Mol Cell 2018 and it is important to note that the protonation state only of histidine changes significantly between 7 and 5.5 and histidine is rare in these sequences), as hnRNPA2 LC phase separates in the presence of salt and NMR signal is broadened beyond detection. We have updated the manuscript to make this clearer.

-

The authors state (in the context of the NMR data in Figure S1E) that hnRNPA2 LC does not phase separate at low salt concentrations, and that hnRNPF PLD is capable of inducing condensation under these conditions. This should be shown by microscopy and the exact conditions (salt, pH, protein conc.) should be given.

We thank the reviewer for this suggestion and have added this data and updated the text to reflect the new finding.

-

Supplemental Figures S1, S2, and S3 are too packed and the displayed images are therefore too small to judge the presence of condensates in some of them. I suggest to split these figures each into 3 separate Figures, and remove most of the TEV- data, since it is obvious from the basic constructs like hnRNPA2 LC +/- TEV that addition of TEV initiates LLPS.

We thank the reviewer for this suggestion and have split the supplemental figures into several more figures.

-

How do you put the changes in hnRNPA2 LC LLPS into context of the previously shown PRMT1 induced methylation of hnRNPA2 and its effects on LLPS? Does any PTM cancel hnRNPA2 LLPS, or are there some that enhance LLPS? please, at least, discuss this.

We thank the reviewer for this suggestion and have added text discussing this to the revised manuscript.

--

If pY is "selectively" canceling LLPS and co-LLPS of others proteins, how about other phosphorylations? Is the presence of phosphate groups per se influencing the granule assembly? You showed pseudo-phosphorylation of serines, but what about using another serine/threonine kinase that actually phosphorylates these residues?

What happens if you phosphorylate the clients hnRNPF PDL or TOG D1, do you see a similar or opposite effect?

We thank the reviewer for this suggestion and we agree it is important. We attempted serine phosphorylating hnRNPA2 LC with casein kinase I (from 2 companies) and casein kinase II as previous work had shown that TDP-43 could be multiply phosphorylated by CKI in vitro and hnRNPA2 could be in vitro phosphorylated by CKII. Unfortunately, were unable to obtain serine phosphorylated hnRNPA2 LC (as determined by mass spectrometry of the purified protein), and

therefore were unable to test the effect of serine phosphorylation. As hnRNPF is also a target of Fyn, we expect tyrosine phosphorylation of hnRNPF could eliminate partitioning into hnRNPA2 droplets -- we have added a comment on this in the revised manuscript.

--

The authors speculate that "hnRNPA2 LC self-interactions between phospho-tyrosine and arginine outcompete the weak interactions with hnRNPF PDL, thus preventing its partitioning." This is an interesting hypothesis that the authors could attempt to test, e.g. by experiments, in which a) hnRNPA2 LC + hnRNPF PDL are mixed to co-condensate, and then pYhnRNPA2 LC is added to see if hnRNPF PDL gets depleted from the condensates. They could also try to outcompete against hnRNPF PDL with soluble pY.

We thank the reviewer for this suggestion. We do not think that adding pY hnRNPA2 LC to mixed hnRNPA2 LC + hnRNPF PLD would be a "clean" experiment. As suggested, we performed mixing experiments in the presence of free amino acids (e.g. pY) and did not find a change in partitioning, though the reason for this is unclear as we did not monitor partitioning of free pY and we note that it is quite different from pY in a chain. As such we have updated the text to reflect that and qualify the language regarding the potential mechanism of arginine interactions.

--

True granules contain more than 2 components; what happens if you mix hnRNPA2, hnRNPF, and TOG D1? Do they compete in partitioning? And how is that regulated by RNA, a major and essential constituent of the granules?

We thank the reviewer for this suggestion and have performed a 3-way mixing experiment between hnRNPA2, hnRNPF, and TOG D1 and find they are all present in droplets at once (Figure 2E and Figure S6 in revised manuscript). We did not include RNA as incorporating appropriately controlled studies with RNA would dramatically increase the scope of the manuscript. As we stated above, the primary goal of the manuscript is to map the interactions that the LC domains make and how these interaction networks are altered by tyrosine phosphorylation. We do acknowledge that RNA can interact with LC domains and it is the topic of other studies in our group outside of the scope of this manuscript.

--

It was observed that glutamines in the positions of tyrosine in hnRNPA2 LC increase the "prion-like" character. Please discuss why that might happen.

We thank the reviewer for this suggestion and have added discussion in the text about this.

--

In addition: across the manuscript, the term "prion-like" is used without definition which characteristics are hiding behind this term. The authors have to at least once in the beginning give a definition what they mean with "prion-like" (LLPS to aggregate transition????) and where this originates from. Real prion-behavior includes structural conversion as well as cell transmission, I guess that's not what you are talking about. I suggest to define or simply describe what is changing instead of naming it "prion-like".

We thank the reviewer for this suggestion and have updated the text accordingly. In our understanding, the reviewer's definition is indeed accurate regarding protein-based-inheritance as the definition of "prion". The term "prion-like" as defined by Shorter/Gilmer/King refers to a sequence that resembles the sequence composition found in yeast prion proteins – and does not refer to phenotype or aggregation/fibril-formation.

It is not defined and explained what PRE and PLAAC are, and what they measure.

We thank the reviewer for catching this oversight and have updated the text.

--

In vivo studies:

Please proof - biochemically for protein and/or RNA levels - the successful knock-down and expression of hrp-1 and Fyn in the C elegans lines.

We thank the reviewer for this suggestion and performed RT-PCR experiments to show that the hrpa-1 chimera is expressed as predicted. We attempted to do RT-PCR for Fyn but we were unable to detect Fyn (methods updated to reflect this fact), likely because Fyn is expressed at low levels in only a few neurons. However, as we have a biological effect of Fyn but not Fynempty, we are confident it is expressed. We did not do this for the hrpa-1 deletion alleles as these deletion alleles either remove most of the 2nd RRM and the LC domain (for ok592) or remove part of RRM1 and introduce a frame shift mutation (for tm781). Both deletion alleles are homozygous lethal/sterile, which matches results from RNAi knockdown studies of hrpa-1 (Fernandez et al 2005 Genome Res; Longman et al 2000 EMBO J among others).*

-

What is the phosphorylation status of HRP-1 with and without expression of Fyn?

We thank the reviewer for this suggestion. Unfortunately, this experiment is almost impossible in C. elegans. Here, Fyn is only expressed in a few neurons and obtaining sufficient HRP-1 from those neurons when it is already difficult to obtain sufficient protein from C. elegans is incredibly difficult.

--

Do the worms show neuronal loss after stress? In Figure S6F, no difference in GFP+ neuronal numbers is shown in absence of stress. What is the number of GFP+ neurons in C elegans after stress application?

We thank the reviewer for this suggestion and have performed this experiment, which showed that there is not substantial neuronal death in hrpa-1(Δ) animals after 22 hours paraquat stress (Figure S11G).

--

Do the lines that carry mScarlet-containing constructs show neurodegeneration or neuronal loss?

We thank the reviewer for this comment. We note that the lines carrying mScarlet constructs express the mScarlet proteins only in mec-4 glutamatergic neurons (which have long processes) and we do not observe neurodegeneration (nor did we expect it). We have updated the text to make this clear and also commented that the spots observed are, therefore, not expected to be toxic assemblies.

--

The experiments showing dye uptake in neurons of WT and D290V HsLCmScarlet lines - presented in Figure 5F+G - seem underpowered with n=3 repetitions. They should be repeated at least 2-3 more times. Additionally, a statistical comparison between stressed and no-stress conditions for all groups is missing.

We thank the reviewer for this suggestion. After discussion with the editor we did not do more trials as these experiments already represent 12 animals per genotype per 3 trials (36 animals total) with more than one neuron scored in most animals.

--

The findings about TDP-43-related neurodegeneration in C elegans are interesting but need a lot more experimental elaboration and - in the current form - seem out of the scope of this manuscript - at least in their current state and as presented: they open a lot of obvious questions about the relationship between hnRNPA2 and TDP43, which are not addressed.

The findings are also not discussed at all, and seem to not contribute much to the "story".

I suggest to take the TDP43 data out from this manuscript and focus on phosphorylation (and other PTMs) and the effect on granule assembly. If left in the ms, the authors should address at least the following questions regarding hnRNPA2:TDP-43 interaction in neurodegeneration: 1) the authors previously showed that hnRNPA2 and TDP43 can co-phase separate; how is co-LLPS effected by tyrosine phosphorylation? 2) Do TDP-43 and hnRNPA2 co-localize in stress-

induced neuronal "spots", in C elegans and in mammalian neurons or at least mammalian neuronal cells? 3) How does Fyn expression alter TDP43 PTMs, phase separation, aggregation, and nuclear:cytosolic localization? It needs to be shown biochemically that TDP-1 is knocked-out in the worms, as well as if the localization of TDP-1 is changed upon stress and Fyn expression, in both the WT and mutant hnRNPA2 LC expressing lines.

We thank the reviewer for this suggestion but after discussion with the editor we have left the TDP-43 data in the manuscript. We note as above that the advantage of the tdp-1 data is that it enables us to further characterize the hnRNPA2 model and its associated neurodegeneration and clearly demonstrate effects of the mutant LC that cannot be accounted for by loss of function. Therefore, tdp-1 data are not intended for comparison with human disease. The reviewer presents a compelling set of experiments that will lead to understanding the mechanistic role of tdp-1 in neurodegeneration, whether or not these interactions are altered in liquid or solid phases, but this is not the focus of this effort and these experiments would dramatically alter the scope of the effort. We have updated the text to highlight the importance of the tdp-1 data in characterizing the hnRNPA2-associated degeneration observed in these novel models.

Minor points:

We thank the reviewer for their helpful and careful reading of the manuscript. We have addressed all the points below, which generally only required small changes to the manuscript text. Below we have only made specific notes for exceptions or if special handling was required.

- hnRNPA2 is multiple times written as hnRNAP2

- Page 2 lane 9: ...these these... > ...these...

- Page 4 lane 1: ... two these > ...these two...

- Page 4 lane 20: ...hnRNPA2 and hnRNPF... > ... hnRNPA2 LC and hnRNPF PLD...

- Page 6 lane 25:of between.... > ...of...

- Page 9 lane 18: ...reduce prion-like... > ...reduce the prion-like....

These typos were fixed.

- Page 5 lanes 1: ... but not known to be > Do you mean "not known to be" or "known to not be"

We edited the text to remove this phrase and hopefully reduce confusion.

- Page 5 lane 20: ..."hnRNPA2 LC-like" patten of charges residues... > do you mean "charge pattern" or "net charge"?

We changed this phrasing to hopefully reduce confusion.

- The authors state specificity of hnPNPF PDL co-condensation with hnRNPA2 LC. However, they also show that exchange of a few charged residues in FUS can enable hnRNPF co-LLPS. -

I would rather interpret this as non-specific co-LLPS of hnRNPF that is mostly dependent on protein charge but not sequence specificity, since the authors showed that hnRNPF PDL can co-condensate with [FUS LC + Rs] because of charge. Later in the manuscript, when tyrosine phosphorylation is analyzed, it is shown that charge change by phosphorylation is "not responsible for specificity of interactions with hnRNPA2 LC" (Page 8). "...not exclusively responsible...." Would be more appropriate here I think. I suggest to be more careful with the chosen phrasing and the interpretation. However, it also seems like a logical conclusion that protein conformations, and thus interactions, are different in the presence of phosphorylated tyrosines (or other residues) compared to glutamates due to the bulky phosphate groups that present three negative charges in close proximity and thereby enable very precise, oriented electrostatic interactions.

We agree with the reviewer that the partitioning is dependent on charged residues (and not on the specific sequence) and have tried to make this clear in the manuscript. We have taken the reviewer's suggestion to say "not exclusively responsible". We agree that protein conformations and the structure of interactions may be different because of the difference in structure of phosphoryl group (and phosphotyrosine specifically) compared to glutamate, but because we do not probe this here we chose not to comment on this interesting possibility.

- In most cases the authors talk about "tyrosine" and "arginine"; this should be changed to "tyrosine residues" or "tyrosine", adequately for arginine and other residues like serine etc. *We have changed these references as suggested.*

- Page 5 lane 22: ..."FUS LC-like" depletion of charged residues...> I suggest to remove the expression "FUS LC-like" or replace like depletion of charged residues, similar to FUS LC, *We edited the text to hopefully increase clarity.*

- Page 6 lane 10: ...we hypothesized that the 1st domain of ch-TOG would also partition into hnRNPA2 droplets. It is unclear what is the rationale behind this hypothesis, in other words where this idea comes from and why not other domains are tested. It would help if the authors would show TOG domain structure and state why D1 likely partitions. *We thank the reviewer for this suggestion. We attempted to both purify FL TOG from a construct from Addgene and clone FL TOG into our own vector and were unsuccessful at both. However, as the individual TOG domains are extremely similar, and previous data in the field suggested that all seven TOG domains have similar interaction with hnRNPA2 (Falkenberg 2017) therefore we felt that using only one was a sufficient model for TOG domain interaction with hnRNPA2.*

- Page 10 lane 15: the authors should give the total number of residues for hnRNPA2 LC and its orthologues and at least mention, better comment on, the reduced number of serines and enhanced number of glutamines in Drosophila and C elegans. *We thank the reviewer for this comment. We have added the number of residues in the LC domains to the figure and commented on the serine/glutamine content in C. elegans.*

- Page 10 lane 17: ...HRP-1 function....> Do the authors mean HRP-1 condensation capability? They are not testing the actually protein function here *We have clarified the text to discuss LLPS/self-assembly rather than function.*

- Page 10 lane 21: please specify "low salt conditions" and "physiological salt (concentration)". - Furthermore, the pH is different between the conditions in reported; does the low pH5.5 lead to aggregation of HRP-1? *We have clarified the salt conditions in the text. We do not think that the lower pH leads to HRP-1 LC aggregation but rather that the increased glutamine content is contributing to make HRP-1 LC act slightly differently than hnRNPA2 LC. Again, we note that these domains are low in histidine residues (which may titrate between pH 5.5 and 7.0) and also in acidic residues (which have a pKa significantly lower than 5.5 in intrinsically disordered regions unless clustered together) (Please note we have changed the nomenclature from "HRP-1" to "HRPA-1" to follow recent changes in C. elegans naming.)*

- Page 10 lane 22/23: ...self-assembly...> what is self-assembly of a LCD? please define more precisely, or use self-association, which is a more broad expression compared to usually well organized self-assembly. *We have changed assembly to association here.*

- Page 10 lane 23: ...we created C.e. strains expressing a chimeric HRP-1 protein containing most of the human hnRNPA2 LC domain. > it is not clear what is the rationale behind these lines; the authors need to explain, what they expect to gain from these lines, and why they chose the chimera as they did.

We have added more text describing the rationale behind the chimeric lines.

- Figure S5A: needs to explain better, which residues of hnRNPA2 LC were transferred and to which location in the hrp-1 LC sequence

We have added an additional table in the appendix giving the full protein sequence of the C. elegans, human, and chimeric HRP-1/hnRNPA2 proteins with the LC and changed residues highlighted.

- Page 11 lane 1: I guess the sentence should be: ...if HRP-1 assembles into stress granules, transport granules, or aggregates, all of them together are referred to herein as "spots".

We updated the text to increase clarity.

- Page 12 lane 2: ...No overt defects were observed... > please specify: morphologically, behaviorally?

We added these qualifiers.

- Page 12 lane 4: ...disease models... > ... disease models of C elegans...

We have clarified that these are C. elegans models.

- Page 13 lane 13: please give name of GFP line please.

This is the osm-10p::GFP line(s). The strain name (see Appendix Table 1) is HA3, the integrated array name is rtls11.

- Page 13 lane 16: it is absolutely unclear what "+" means the text. This is also true in the corresponding figure panels. The authors need to clearly state which line or treatment is labeled with "+".

We have added a sentence describing the difference between WT and + (+ is a WT animal whose mother was WT/balancer to control for the necessity of the balancer in the genetic background).

- Page 13, paragraph starting lane 24: It seems to me that this paragraph would have a better place further up, right after the authors report the data from the mScarlet constructs in the context of Fyn?

We thank the reviewer for this suggestion; however we left the paragraph in place because we have not established neurodegeneration in C. elegans when we first discuss Fyn with the mScarlet lines.

- Page 14 lane 11 (discussion part): ...distinct prion-like domains. > I suggest to change to distinct low-complexity domains.

We made this change.

- Page 15 lane 16: ...assemblages... > ...assemblies...

We made this change.

- Page 15 lane 19: can you comment on the role of Fyn in other neurodegenerative diseases?

Would it also make sense to boost Fyn in other disease contexts?

We added some discussion of this point.

Figure 1:

- Add protein domain and construct schematic for hnRNPA2, hnRNPF, and FUS

We added these domain constructs schematics.

- The data for the FUS control in (B) could go in supplemental and instead show the quantification as in Figure S1J)

As this is the first time we show FUS LC LLPS in this paper, we left the FUS/hnRNPF control images.

- Add imaging data for FUS LC(40uM)+ hnRNPF PDL (40uM) (no co-LLPS?) to panel D. Also, for easy readability, please give protein concentrations in all figure panels.

We did not perform this experiment as there is no LLPS of FUS LC at 40 μ M.

- Panel c): unclear what the different colored bars are; please describe in the legend. What is "physiological pH"? What are the negatively charged residues?

We have changed "physiological" to "neutral". The negatively charged residues are now indicated.

Figure 3:

- Panel c): nomenclature different than in ms text: pYhnRNPA2 instead of phnRNPA2

We thank the reviewer for catching this error and have fixed the figure.

Figure 4:

- Panel b): what is PLAAC and what does it measure?

Figure 5:

- Panel d): which worm genotype is shown?

These images were from a variety of worm genotypes to demonstrate the range of spot distribution observed. The figure legend has been updated to reflect that fact.

- Panel e): marker size in graph are not uniform

The markers are now uniform.

- Panel f+g): how many animals and neurons did you analyze? From the data point distribution and SD it looks like these experiments are underpowered and more animals and/or repetitions of the experiment have to be added. Also problematic: the way that the conditions are compared. why is the HsLcWT without Fyn not giving a difference between stress and no-stress as before reported in panel E)? Were those conditions statistically compared at all in this experiment?

There were 12 animals/trial/genotype scored. There were 1-4 neurons scored per animal (depending on mosaic expression of the transgenes and body posture of the immobilized animal). Most animals had 2 neurons counted. We did not compare HsLcWT in panels F and G between stress and no stress conditions because the animals were not scored at the same time.

Figure 7:

- The figure legend is poor. What are the red crosses standing for? In the in vivo part, what is this supposed to be?

We have updated the figure and legend to hopefully address these points.

Figure S1:

- Figure is too crowded, too small images. Please split Figure in 2-3 figures: e.g. A+B; C+D+E+F; G+H+I+J
- Same is true for Figure S2 and S3

We thank the reviewer for this suggestion and have split these supplemental figures into many.

Figure S6:

- Panel F: what does "ASH" mean? What about the number of GFP+ neurons in stressed animals?

We performed the assay in stressed animals. ASH is the name of a C. elegans neuron which has been added to the figure legend.

Additional comments:

The authors use an overly complicated nomenclature of the used constructs; I suggest to find another shorter but intuitive way to name the constructs, e.g. "hnRNPA2*" instead of "hnRNPA2 LC no charge" and "hnRNPA2*R" instead of "hnRNPA2 LC no charge with R", or similar; R to K could be written as R/K. This will make the text much better readable and understandable.

*We use * elsewhere in the manuscript to mean something else.*

We changed the name of these constructs to hopefully be more clear: "hnRNPA2 LC no charge" is "hnRNPA2 LC^{CD}" where "CD" is defined as "charge depleted"; "hnRNPA2 RtoK" is now "hnRNPA2 LC^{R→K}", etc.

--

At many places in the manuscript, the order of Figure panel is not in order; this has to be corrected by either text modification or Figure rearrangement.

We have checked the figure panel order and reference them all in order now.

--

Often poor figure legends that lack description of experimental setup, conditions, statistics
These have been edited to increase clarity.

Dear Nick,

Thanks for submitting your revised manuscript to the EMBO Journal. Your manuscript has now been re-reviewed by referee #2 and the comments are provided below.

As you can see from the comments the referee appreciates the introduced changes, but also find that some of the previously raised points have not been adequately addressed. I suspect that you should be able to address points 1-3, and 6. Point #4 is concerning the phosphorylation state of hnRNPA2 in cells and if it is phosphorylated by Fyn. I see where the referee is coming from and having such data would strengthen the findings. What would it take to obtain such data? Regarding point #5, I think this issue can be clarified with a more careful description of the data and what you show and don't.

Would be good to discuss everything further.

With best wishes

Karin

Karin Dumstrei, PhD
Senior Editor
The EMBO Journal

The revision must be submitted online within 90 days; please click on the link below to submit the revision online before 18th Nov 2020.

Referee #2:

In the manuscript "Tyrosine phosphorylation regulates hnRNPA2 granule protein partitioning & reduces neurodegeneration", Ryan et al. use recombinant proteins to describe the co-partitioning of domains of the transport granule components hnRNPF and TOG into condensates formed by the hnRNPA2 low complexity domain (hnRNPA2 LC), that co-partitioning depends in part on charge and charge distribution in the hnRNPA2 LC, and how the condensation behavior of hnRNPA2 LC (and two disease mutants) is decreased by phosphorylation on tyrosine residues. Full-length proteins are also examined. By NMR they identify the interaction of soluble non-phase separated hnRNPA2 LC with hnRNPF PLD and TOG D1. To show that hnRNPA2 granules also form in *C. elegans* in vivo, they express mScarlet-tagged hnRNPA2 or mScarlet-tagged chimeric hnRNPA2 containing the RNA-binding domains of *C. elegans* orthologue hrpa-1 fused to hnRNPA2 LC. Using these lines they show that oxidative stress induces hnRNPA2 granules, that mutant hnRNPA2 increases the number of granules, and that co-expression of Fyn kinase decreases the amount of granules, suggesting that tyrosine phosphorylation by Fyn inhibits hnRNPA2 LLPS in vitro and granule formation in living worms. Mutant hnRNPA2 induces neurodegeneration under oxidative stress, which is rescued by Fyn expression and tdp-1 knockdown. The latter is interpreted as tdp-1 being a transmitter of mutant hnRNPA2 induced neurodegeneration.

The manuscript has improved by the changes requested and addressed in the last revision round - especially the figures are more clear now and missing information has been added. However, I think several important issues stated by the reviewers could not be resolved, which still largely compromise the quality of the manuscript.

Major points:

1. The image quality remains poor for multiple conditions tested, in a way that the presented images do not support the data interpretation; images in Figs 2E, Fig 3D, and Figs S1, S3, S4, S6, S8.
2. There seem to be big differences between the condensation of hnRNPA2 in etc different conditions, and the amount of phase separation, e.g. the number of droplets or area covered) could easily be assessed, however remains to be not determined. As an example, in Figure S1B, the

amount of LLPS in hnRNPA2+TEV with and without hnRNPF PLD/DeltaPLD is extremely different. This should be noted in the text. Were the images taken at the same conditions? and after which incubation times? Another example: hnRNPF DeltaPLD+TEV looks very different in Figure S1C and B; how can that be?

3. The co-partitioning of TOG into liquid-like droplets containing hnRNPA2 and hnRNPF cannot be concluded from the presented images; it rather looks like aggregation to me. Better images needed.

4. It remains unresolved a) what is the phosphorylation state of hnRNPA2 in cells, and if hnRNPA2 (or its LCD) gets phosphorylated by Fyn. This could quite easily be addressed in cells, including the effect of stress on hnRNPA2 phosphorylation, which seems to be needed since the authors stated that the amount of hnRNPA2 from (stressed) worms was too little to be analyzed. Since the authors try to connect their findings with the occurrence of hnRNPA2 granules in neurons and with aggregation in disease, using mammalian cultured neurons or neuronal cell lines for these experiments might have been desirable. In fact, it might have been more straight forward - and maybe also informative - to use mammalian cells in culture than utilizing *C. elegans* for this study, which here cannot unravel its full power as an *in vivo* behavioral and imaging model, but rather complicates the examination of mammalian stress granule biology.

5. The data on *tdp-1* in *C. elegans* seem out of place and rather confusing than relevant for the presented story. On one hand the authors claim in the abstract that mutant hnRNPA2 neurodegeneration in worms depends on TDP-43; this has not been tested, and the authors replied to the reviewers that they don't want to compare *tdp-1* with TDP-43, yet they claim it in the manuscript without testing the effect of TDP-43 at any point. The effect of Fyn on *tdp-1* dependent hnRNPA2 toxicity remains untested as well, and no mechanism is suggested.

6. >>hnRNPF PLD did not partition into hnRNPA2 LCcd but did partition into FUS LCce (Figure 1D, Appendix Figure S3B), consistent with our hypothesis.<< - I see this statement problematic since the authors used different concentrations of hnRNPF PLD together with hnRNPA2 LCcd and FUS LCce. Furthermore, Figure 1D shows LLPS of FUS LCce, but in Figure S3B there is no phase separation of FUS LCce, and therefore also no co-partitioning. Where does this discrepancy come from?

Response to reviewers round 2

Referee #2:

In the manuscript "Tyrosine phosphorylation regulates hnRNPA2 granule protein partitioning & reduces neurodegeneration", Ryan et al. use recombinant proteins to describe the co-partitioning of domains of the transport granule components hnRNPF and TOG into condensates formed by the hnRNPA2 low complexity domain (hnRNPA2 LC), that co-partitioning depends in part on charge and charge distribution in the hnRNPA2 LC, and how the condensation behavior of hnRNPA2 LC (and two disease mutants) is decreased by phosphorylation on tyrosine residues. Full-length proteins are also examined. By NMR they identify the interaction of soluble non-phase separated hnRNPA2 LC with hnRNPF PLD and TOG D1. To show that hnRNPA2 granules also form in *C. elegans* in vivo, they express mScarlet-tagged hnRNPA2 or mScarlet-tagged chimeric hnRNPA2 containing the RNA-binding domains of *C. elegans* orthologue hrpa-1 fused to hnRNPA2 LC. Using these lines they show that oxidative stress induces hnRNPA2 granules, that mutant hnRNPA2 increases the number of granules, and that co-expression of Fyn kinase decreases the amount of granules, suggesting that tyrosine phosphorylation by Fyn inhibits hnRNPA2 LLPS in vitro and granule formation in living worms. Mutant hnRNPA2 induces neurodegeneration under oxidative stress, which is rescued by Fyn expression and tdp-1 knockdown. The latter is interpreted as tdp-1 being a transmitter of mutant hnRNPA2 induced neurodegeneration.

The manuscript has improved by the changes requested and addressed in the last revision round - especially the figures are more clear now and missing information has been added. However, I think several important issues stated by the reviewers could not be resolved, which still largely compromise the quality of the manuscript.

Major points:

1. The image quality remains poor for multiple conditions tested, in a way that the presented images do not support the data interpretation; images in Figs 2E, Fig 3D, and Figs S1, S3, S4, S6, S8.

We have moved some supplemental images from the main text (2E) and updated figure captions to address this question (3D). For Figure S1B we replaced the bottom panel with a different region of the image that shows more droplets. We also note that we perform droplet microscopy experiments in the absence of crowding agents or large protein tags, unlike others in the field, which have unknown effects on the biochemistry and biophysics of the system but dramatically (and perhaps non-physiologically) enhance LLPS and may make images appear more uniform due to the presence of more droplets. We choose not to use crowding agents whenever possible. We also perform these assays at low micromolar protein concentrations, which makes the droplets more spread out in the field of view, which we have found improves image quality of the droplets imaged (less likely to have droplets in different planes simultaneously, less likely to have droplets coalesce in the time it takes for the camera to image, etc.). We disagree that the images presented do not support the interpretation as we do not make quantitative statements about these images, only qualitative statements of partitioning vs. no partitioning.

2. There seem to be big differences between the condensation of hnRNPA2 in etc different conditions, and the amount of phase separation, e.g. the number of droplets or area covered) could easily be assessed, however remains to be not determined. As an example, in Figure S1B, the amount of LLPS in hnRNPA2+TEV with and without hnRNPF PLD/DeltaPLD is extremely different. This should be noted in the text. Were the images taken at the same

conditions? and after which incubation times? Another example: hnRNPF DeltaPLD+TEV looks very different in Figure S1C and B; how can that be?

Where possible, we show differences in hnRNPA2 LLPS with different mutations by the measuring the concentration of the protein remaining in the supernatant after centrifugation, a partitioning assay that readily lends itself to quantification (see Figures 1F, 3A, and S7B). We disagree that image quantification is the right way to test these differences, due to the effect of uncontrollable variables (room temperature or humidity, time exposed to air on slide, time required to focus on droplets and image, etc.) on droplet size and quantity in our microscopy assays. The longer the droplets remain on the slide the larger they become due to droplet coalescence (see figure 4A for example of images taken at different time points showing increased droplet size with time) and small amounts of evaporation of water, thus increasing the protein concentration in the solution. For the specific examples the reviewer points to, Figure S1B we have now changed the images for hnRNPA2 FL + TEV + hnRNPF PLD to show more droplets (the original image is from a different region of the same microscopy image and is a region showing few droplets but good signal in the red channel); for the difference between hnRNPF Δ PLD in Figure S1B and C, these experiments were done at different buffer conditions (indicated in the supplemental figure legends and discussed in the results section in order to compare to the conditions used in the different experiments) which explains the apparent discrepancy.

3. The co-partitioning of TOG into liquid-like droplets containing hnRNPA2 and hnRNPF cannot be concluded from the presented images; it rather looks like aggregation to me. Better images needed.

We agree with the reviewer that we should more carefully present and discuss this section. The images in Figure 2E do show colocalization of hnRNPA2 LC, hnRNPF PLD, and TOG, but these do indeed appear irregularly shaped (as we describe elsewhere in the manuscript, hnRNPF PLD appears to reduce fluidity (or cause aggregation) of hnRNPA2 LC droplets). Therefore, we have added images previously from the SI to the main text to address this concern where we show that hnRNPF full-length and TOG D1 co-localize simultaneously to round granules. Please also see response to point 1.

4. It remains unresolved a) what is the phosphorylation state of hnRNPA2 in cells, and if hnRNPA2 (or its LCD) gets phosphorylated by Fyn. This could quite easily been addressed in cells, including the effect of stress on hnRNPA2 phosphorylation, which seems to be needed since the authors stated that the amount of hnRNPA2 from (stressed) worms was too little to be analyzed. Since the authors try to connect their findings with the occurrence of hnRNPA2 granules in neurons and with aggregation in disease, using mammalian cultured neurons or neuronal cell lines for these experiments might have been desirable. In fact, it might have been more straight forward - and maybe also informative - to use mammalian cells in culture than utilizing *C. elegans* for this study, which here cannot unravel its full power as an in vivo behavioral and imaging model, but rather complicates the examination of mammalian stress granule biology.

*We recognize the limitations of *C. elegans* to provide biochemical detail in this case but we do want to point to the work that was previously done by White et al. (2008) Activation of oligodendroglial Fyn kinase enhances translation of mRNAs transported in hnRNP A2-dependent RNA granules. J Cell Biol 181: 579-586 in mammalian cells on hnRNPA2 phosphorylation. They “identify hnRNP A2 as a target of activated Fyn”, “show that active Fyn phosphorylates hnRNP A2 and stimulates translation of an MBP A2RE-containing reporter construct”, and that “Fyn activation... leads to an increase in hnRNP A2 phosphorylation.” While the reviewer’s proposed series of experiments are certainly interesting and will help for understanding transport granule biology in great detail, we feel that performing these*

experiments are beyond the scope of the current manuscript as it would require a novel collaboration, take a large amount of time, and mammalian cell culture experiments will not tell us what is happening in the C. elegans neurons assayed. This last point is most important – if we would see that hnRNPA2 is not phosphorylated in neuronal culture, that does not tell us it is not happening in the Fyn rescue in C. elegans. Yet, we already know from the literature (White et al.) that activated Fyn phosphorylates hnRNPA2.

5. The data on tdp-1 in C elegans seem out of place and rather confusing than relevant for the presented story. On one hand the authors claim in the abstract that mutant hnRNPA2 neurodegeneration in worms depends on TDP-43; this has not been tested, and the authors replied to the reviewers that they don't want to compare tdp-1 with TDP-43, yet they claim it in the manuscript without testing the effect of TDP-43 at any point. The effect of Fyn on tdp-1 dependent hnRNPA2 toxicity remains untested as well, and no mechanism is suggested. *We have updated the manuscript text to make it very clear that we do not claim that neurodegeneration depends on TDP-43, rather the ortholog of TDP-43, tdp-1. We have also reworked this section to state clearly the main finding that tdp-1 data provides evidence for gain-of-function contributions in the C elegans model yet preserving our inspiration to test tdp-1 due to the biochemical and pathological connections. We have removed TDP-43 from the abstract. We have also added caveats in the section of the results section regarding the relevance to human disease. Additionally, we state in the discussion “Conversely, our results also demonstrate that the toxicity caused by mutant hnRNPA2 expression, but not toxicity caused by loss of the C. elegans ortholog of hnRNPA2, is ameliorated by loss of the ortholog of TDP-43 (Figure 7), a related protein that forms aggregates in the majority of ALS cases (Mackenzie & Rademakers, 2008) and co-aggregates with hnRNPA2 in MSP (Kim et al., 2013). Therefore, future work further elucidating how these proteins interact genetically, physiologically, and pathologically will be important for evaluating their combined contribution to neurodegeneration.” We cannot evaluate the effect of Fyn on tdp-1 dependent hnRNPA2 toxicity using our genetic model because tdp-1 deletion leads to rescue of degeneration as does the expression of Fyn.*

6. *>>hnRNPF PLD did not partition into hnRNPA2 LCcd but did partition into FUS LCce (Figure 1D, Appendix Figure S3B), consistent with our hypothesis.<< - I see this statement problematic since the authors used different concentrations of hnRNPF PLD together with hnRNPA2 LCcd and FUS LCce. Furthermore, Figure 1D shows LLPS of FUS LCce, but in Figure S3B there is no phase separation of FUS LCce, and therefore also no co-partitioning. Where does this discrepancy come from?*

This difference in occurrence of phase separation is due to a difference in protein concentration used to prepare droplets for the images indicated. FUS LC^{CE} does not phase separate at 20 μ M (in the supplement) but does at 40 μ M (supplement top image and main text image). We have changed the labels on the figures and the figure legend to clarify these differences.

Dear Nick,

Thank you for submitting your revised manuscript to The EMBO Journal. I have now had a chance to take a look at everything and I appreciate the introduced changes. I am therefore very pleased to accept the manuscript.

Congratulations on a nice study!

Best Karin

Karin Dumstrei, PhD
Senior Editor
The EMBO Journal

Please note that it is EMBO Journal policy for the transcript of the editorial process (containing referee reports and your response letter) to be published as an online supplement to each paper. If you do NOT want this, you will need to inform the Editorial Office via email immediately. More information is available here: https://emboj.embopress.org/about#Transparent_Process

Your manuscript will be processed for publication in the journal by EMBO Press. Manuscripts in the PDF and electronic editions of The EMBO Journal will be copy edited, and you will be provided with page proofs prior to publication. Please note that supplementary information is not included in the proofs.

Should you be planning a Press Release on your article, please get in contact with embojournal@wiley.com as early as possible, in order to coordinate publication and release dates.

If you have any questions, please do not hesitate to call or email the Editorial Office. Thank you for your contribution to The EMBO Journal.

Corresponding Author Name: Nicolas L. Fawzi and Anne C. Hart

Manuscript Number: EMBOJ-2020-105001R